# Cortical state dynamics and selective attention define the spatial pattern of correlated variability in neocortex

Yan-Liang Shi [1], Nicholas A. Steinmetz [2], Tirin Moore [3,4], Kwabena Boahen [5,6] & Tatiana A. Engel [1] ✉

Correlated activity fluctuations in the neocortex influence sensory responses and behavior. Neural correlations reflect anatomical connectivity but also change dynamically with cognitive states such as attention. Yet, the network mechanisms defining the population structure of correlations remain unknown. We measured correlations within columns in the visual cortex. We show that the magnitude of correlations, their attentional modulation, and dependence on lateral distance are explained by columnar On-Off dynamics, which are synchronous activity fluctuations reflecting cortical state. We developed a network model in which the On-Off dynamics propagate across nearby columns generating spatial correlations with the extent controlled by attentional inputs. This mechanism, unlike previous proposals, predicts spatially non-uniform changes in correlations during attention. We confirm this prediction in our columnar recordings by showing that in superficial layers the largest changes in correlations occur at intermediate lateral distances. Our results reveal how spatially structured patterns of correlated variability emerge through interactions of cortical state dynamics, anatomical connectivity, and attention.

[1] Cold Spring Harbor Laboratory, Cold Spring Harbor, NY, USA. [2] Department of Biological Structure, University of Washington, Seattle, WA, USA.
[3] Department of Neurobiology, Stanford University, Stanford, CA, USA. [4] Howard Hughes Medical Institute, Stanford University, Stanford, CA, USA.
[5] Department of Bioengineering, Stanford University, Stanford, CA, USA. [6] Department of Electrical Engineering, Stanford University, Stanford, CA, USA.
✉email: engel@cshl.edu

Neocortical circuits spontaneously generate varying patterns of neural activity, which profoundly influence sensory responses and behavior[1–5]. These endogenous activity fluctuations are correlated across neural populations and are often quantified by correlations between pairs of neurons, called noise correlations[6]. Noise correlations are thought to reflect the anatomical circuit connectivity, but they are also dynamically influenced by behavioral and cognitive states[5,7–9], in particular, during spatial attention[10–14]. Implications of noise correlations for population coding and behavior have been studied extensively[15–19]. Yet, how anatomical connectivity and cognitive states interact to define the structure of noise correlations across populations is not well understood.

Spatial selective attention offers a rich experimental domain for studying the combined influence of anatomical connectivity and cognitive factors on the population structure of noise correlations. Changes in noise correlations during attention have been measured across different anatomical dimensions, yielding heterogeneous results. Many studies of noise correlations involved recordings from neurons in different cortical columns, e.g., using rectangular Utah arrays which preferentially sample from laterally separated neurons in more superficial cortical layers[10,12] (Fig. 1a). These studies found that noise correlations substantially decreased when attention was directed to the receptive fields (RFs) of recorded neurons[10–12]. More recent studies used linear multi-electrode arrays to measure attentional modulation of noise correlations within cortical columns (Fig. 1a) and found effects that varied with layer and area. In V4, noise correlations decreased during attention only in input layers during stimulus-evoked but not spontaneous activity, and no significant changes were observed in superficial and deep layers[13]. In V1, noise correlations decreased only in supragranular layers with no significant changes in granular and infragranular layers[14]. In both areas, the magnitude of changes in noise correlations within columns appeared an order of magnitude smaller compared to a sizable reduction of correlations across columns. These data suggest that attentional modulation in correlated variability is not uniform across anatomical dimensions, but depends on lateral distance and cortical layer. The network mechanisms underlying these heterogeneous modulations are unknown.

We hypothesized that heterogeneous changes in noise correlations arise from the modulation of On-Off dynamics propagating locally across columns. The On-Off dynamics are spontaneous transitions between phases of vigorous (On) and faint (Off) spiking that occur synchronously across layers of neocortex[20,21] and are observed in visual cortex of behaving monkeys[22,23] (Fig. 1b). The On and Off phases of population activity persist on the timescale of ~100 ms with exponentially distributed durations[22,23] and correlate with large fluctuations in neurons' membrane potentials[24,25], which are signatures of bistable dynamics[26]. The On-Off dynamics reflect the global cortical state associated with arousal and are also modulated locally within retinotopic maps during selective attention[22,23]. We analyzed spiking activity recorded within cortical columns in V4[27] and found that the scale of On-Off dynamics predicted the magnitude of noise correlations and their dependence on lateral distance.

To explain the spatial patterns of noise correlations, we developed a network model of interacting columns with spatially structured connectivity. The key mechanism in our model is On-Off dynamics that propagate across columns to form spatiotemporal population activity which shapes the structure of noise correlations. Cortical activity propagates laterally as waves across different spatial scales, brain states and behavioral conditions[28,29], and wave-like propagation of spontaneous activity fluctuations is observed in the visual cortex of behaving primates[30]. In our model, attentional inputs shift the stability of local On-Off dynamics, which effectively regulates the efficacy of lateral interactions among columns and affects the spatial activity propagation. As a result, attentional inputs reduce the spatial extent of On-Off dynamics, leading to spatially non-uniform changes in noise correlations. This mechanism predicts that, during attention, noise correlations can change substantially at intermediate lateral distances (across columns) even when average changes within columns are very small. To test this prediction, we analyzed how changes in noise correlations depend on lateral distance in columnar recordings in V4. While average changes were small, when sorted by the lateral distance, the changes in noise correlations were near zero at zero distance and gradually increased with distance in superficial layers, consistent with predictions of our model.

The mechanism based on bistable On-Off dynamics differs from previous models of cortical variability operating in a balanced excitatory-inhibitory regime, where population activity fluctuates around a single global fixed point[31,32]. Balanced networks with a global fixed point do not capture the slow timescale and bistable characteristics of cortical fluctuations that we observed in our data. In addition, the previous model predicts that attentional changes of noise correlations are spatially uniform[31], in opposition to our experimental observations. Our results indicate that visual cortex operates in a regime of local bistable dynamics in single columns that interact via structured anatomical connectivity. Our work provides a unifying framework that explains how heterogeneous patterns of correlated variability emerge within neocortex through interactions of network dynamics and cognitive state.

## Results

**On-Off dynamics predict the magnitude of noise correlations.** We measured spiking activity from all layers within columns of the visual cortical area V4[27]. Spiking activity was recorded with 16-channel linear array microelectrodes (Fig. 1a) arranged so that receptive fields (RFs) on all channels largely overlapped[22,33]. During recordings, monkeys performed a spatial attention task, which required detecting changes in the orientation of a visual stimulus in the presence of distractor stimuli (Methods section and Supplementary Fig. 1). On each trial, an attention cue indicated the stimulus that was most likely to change. In the attention condition, the cue directed animal's attention to the RF stimulus. In the control condition, the cue directed attention to a location outside the RFs of recorded neurons.

In our columnar recordings, we examined the relationship between the scale of ongoing On-Off dynamics and the magnitude of noise correlations. We quantified the On-Off dynamics by fitting a two-phase Hidden Markov Model (HMM) to the population spiking activity[22,23] (Fig. 1b, Methods section). The HMM models the dynamics of a latent population state that switches between two phases, On and Off, to capture synchronized changes in firing rates across neurons. Spikes on recorded channels were modeled as inhomogeneous Poisson processes with different mean rates during the On and Off phases. The variance explained by a two-phase HMM ($R^2$) varied across recording sessions and this variation was tightly correlated with the average noise correlation (Fig. 1c). For most recording sessions (31 total, 67%), the two-phase HMM was the most parsimonious model among HMMs with 1 or up to 8 possible phases[22] (Methods section). For the remaining 15 (33%) sessions, a one-phase HMM (i.e. constant firing rates without On-Off transitions) was the most parsimonious model. These one-phase recordings consistently showed lower average noise correlations of multi-unit (MU) activity (mean 0.13) than two-phase recordings (mean

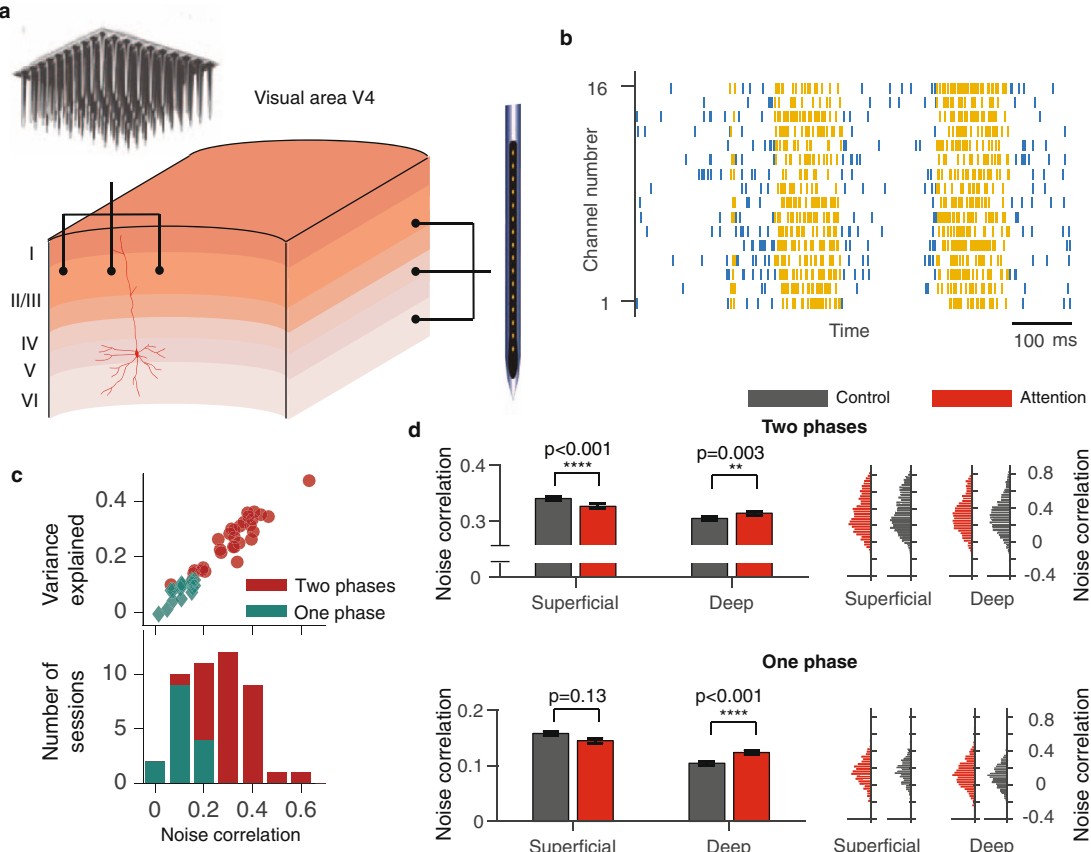

**Fig. 1 On-Off dynamics predict the magnitude of noise correlations within cortical columns. a** Different recording techniques sample neurons along different anatomical dimensions. A rectangular Utah multi-electrode array (top) samples from laterally separated neurons in different columns, preferentially from superficial cortical layers. A linear multi-electrode array (right) samples from neurons across all layers within cortical columns. **b** An example trial showing spontaneous transitions between episodes of vigorous (On) and faint (Off) spiking in multi-unit activity simultaneously recorded from all layers of a cortical column in V4. Spikes (vertical ticks) on 16 recording channels are segmented into On (yellow) and Off (blue) episodes by the HMM. **c** Scatter plot of the variance explained by the two-phase HMM versus average noise correlation of MU across recording sessions for two-phase (red circles) and one-phase (teal diamonds) recordings (upper panel). Two-phase HMM accounted for a smaller fraction of total variance in one-phase compared to two-phase recordings, because in one-phase recordings, the firing-rate fluctuations were smaller and most of the total variance was due to unpredictable Poisson variability. Stacked histogram of average noise correlations for two-phase (red) and one-phase (teal) recordings (lower panel). **d** Average noise correlations of MU in two-phase (upper panels) and one-phase (lower panels) recordings, separately for superficial (two-phase: $n = 2544$ MU pairs for each attention condition; one-phase: $n = 920$) and deep cortical layers (two-phase: $n = 3064$ MU pairs for each attention condition; one-phase: $n = 1448$), in attention (red) and control (gray) conditions (left panels). Histograms show the corresponding distributions of noise correlations in each condition (right panels). Noise correlations slightly decrease in superficial and increase in deep layers, but the overall magnitude of changes is very small. $p$-values are from two-sided $t$-test (two-phase: $p = 2 \times 10^{-5}$ in superficial layers, $p = 0.003$ in deep layers; one-phase: $p = 0.13$ in superficial layers, $p = 4 \times 10^{-7}$ in deep layers). Error bars represent SEM. Source data are provided as a Source Data file.

0.32), with a pronounced 59% difference on average (Fig. 1c). Trial-to-trial variability of MU activity, quantified by the Fano factor (FF, the ratio of the spike-count variance to the mean), was also lower in one-phase (mean FF = 1.5) compared to two-phase (mean FF = 2.3) recordings (35% difference). On the other hand, the mean firing rates of MUs were similar between the one-phase (108 Hz) and two-phase (114 Hz) recordings (5% difference). Thus the scale of On-Off dynamics predicted the overall magnitude of correlated variability in our data, which implicates On-Off dynamics as a major source of noise correlations in visual cortex.

**Attentional modulation of noise correlations within columns.** We quantified attention-related changes in noise correlations within columns, separately in superficial (which included granular and supragranular) and deep (infragranular) cortical layers. In each session, data from each of the recording channels were assigned laminar depth, relative to a common current source

density marker[33]. We combined the granular and supragranular layers because they showed similar changes in noise correlations (Supplementary Fig. 2). We found that noise correlations were slightly reduced in superficial and enhanced in deep layers in the attention relative to control conditions (Fig. 1d). To quantify these changes, we calculated a standard modulation index $MI_{corr}$, which was the difference between noise correlations in the attention and control conditions divided by the sum. In two-phase recordings, the mean $MI_{corr}^{MU}$ for MU was $-0.029$ in superficial layers ($p < 10^{-5}$, Wilcoxon signed-rank test, $n = 5088$) and $0.022$ in deep layers ($p = 0.004$, Wilcoxon signed-rank test, $n = 6128$) (see Supplementary Tables 1 and 2 and Supplementary Fig. 3 for a full summary of results). The magnitude and laminar profile (Supplementary Fig. 2) of these noise-correlation changes are consistent with other laminar recordings in V4[13].

These average changes of noise-correlations within columns were much smaller than the robust and sizable reduction of noise correlations previously reported for neurons in different

columns[10–12,31]. For comparison, a previous study in V4 using rectangular Utah arrays[10] found that the mean $MI_{corr}^{MU}$ was −0.29, which is an order of magnitude larger than in our columnar recordings. Despite this striking difference in the modulation of noise correlations, the attentional modulation of firing rates and trial-to-trial variability of individual neurons was similar in our data and the previous study. In two-phase recordings, the mean $MI_{rate}^{MU}$ was 0.023 in superficial layers ($p < 10^{-10}$, Wilcoxon signed-rank test, $n = 1752$) and 0.018 in deep layers ($p < 10^{-10}$, Wilcoxon signed-rank test, $n = 2216$), which is more comparable to the previous study[10]. Similarly, in two-phase recordings, the mean modulation index of Fano factor $MI_{FF}^{MU}$ was −0.010 in superficial layers ($p < 10^{-10}$, Wilcoxon signed-rank test, $n = 1752$) and −0.007 in deep layers ($p < 10^{-10}$, Wilcoxon signed-rank test, $n = 2216$), comparable to the previous study[10]. The results were similar in one-phase recordings (Supplementary Tables 1 and 2). These results suggest that attention-related changes in noise-correlations depend on the relative positions of neurons in the cortex, with sizable changes across columns and minute, layer-dependent changes within columns. Since the striking difference in the modulation of noise correlations is not accounted for by differences in the activity of individual neurons, it likely reflects the spatial structure of population dynamics across the cortex.

**Network model of interacting cortical columns.** We hypothesized that heterogeneous modulations of noise-correlations across layers and columns arise from the On-Off population dynamics and spatial structure of anatomical connectivity in the cortex. To test this hypothesis, we developed a network model of interacting columns with spatially structured connectivity (Fig. 2). The model consists of units interconnected in two parallel two-dimensional lattices, corresponding to the superficial and deep cortical layers (Fig. 2a). Each unit represents a local population of neurons within one layer—superficial or deep—of a single column. Each unit is connected to its four neighboring units in the same layer, mimicking the local structure of horizontal connectivity in the cortex. Visual stimuli and attention are modeled by external inputs to local groups of units. Since attentional modulation differs between superficial and deeper layers, we modeled each layer as a separate network. Each network receives different attentional inputs to account for differential changes in noise correlations in superficial versus deep layers.

The key mechanism generating correlated variability in our model is the stochastic On-Off dynamics of the population activity in single columns. In visual cortex of behaving monkeys, the durations of On and Off episodes are distributed exponentially with a timescale of ~100 ms[22,23] (Supplementary Fig. 4), which indicates that the On and Off phases are metastable with transitions driven by noise[26]. Accordingly, we model the dynamics of each unit in the network by a two-dimensional dynamical system with two stable fixed points, corresponding to the On and Off phases (Fig. 2b). This dynamical system is a phenomenological mean-field description of a population of excitatory neurons coupled by the vertical recurrent connectivity within the column. The first dynamical variable $r(t)$ represents the mean firing rate of the population. It receives a recurrent self-coupling $F(r)$ and a negative feedback from the second dynamical variable $a(t)$ representing firing-rate adaptation[21,26]. The dynamical system is driven by white noise $\xi(t)$, which causes stochastic transitions between the On and Off fixed points. Each unit also receives external currents $I_{stim}(t)$ and $I_{att}(t)$, which model the bottom-up inputs from visual stimuli and top-down inputs during attention, respectively.

The dynamics of individual units reproduce the On-Off transitions in single columns and their modulation during attention. As in the data from visual cortex[22,23] (Supplementary Fig. 4), the duration of On and Off episodes in the model are irregular and exponentially distributed (Fig. 2c). In this regime, the dynamics of each unit can be reduced to a two-state Markov process switching between the On and Off phases (Supplementary Note 2.1). The Off-to-On ($\alpha_1$) and On-to-Off ($\alpha_2$) transition rates of the Markov process set the average duration of the On and Off episodes: $\tau_{on} = 1/\alpha_2$ and $\tau_{off} = 1/\alpha_1$. Consistent with this description, a two-state HMM provides the most parsimonious fit of the On-Off dynamics in our two-phase recordings. Further, our model captures the increase of On-episode durations during attention as observed in the data[22,23] (Supplementary Fig. 5). During attention, a local group of units representing the attended RFs receives a small excitatory input $I_{att}$. This attentional input slightly shifts the r-nullcline (Fig. 2b) elevating the threshold for transitioning from the On to Off fixed point, which reduces the On-to-Off transition rate of the Markov process and results in longer average On-episode durations (Fig. 2c).

The horizontal connectivity in the network correlates the On-Off dynamics across units in the lateral dimension. Each unit's firing-rate variable $r(t)$ receives a recurrent excitatory input $I_{rec}$ from its four neighbors on the lattice. As a result, the On-Off dynamics of each unit are influenced by the activity of its neighbors. The more neighbors in the On phase, the larger is the excitatory input $I_{rec}$, which elevates the threshold for On-to-Off and lowers the threshold for Off-to-On transitions. In the description of a two-state Markov process, this is equivalent to a dependence of the Off-to-On and On-to-Off transition rates on the On/Off phases of the neighbors: $\alpha_1 + \beta S_{\pm}$ and $\alpha_2 - \beta S_{\pm}$ (Supplementary Note 2.3). Here, the variable $S_{\pm}$ indicates the number of neighbors in the On phase at each time, and $\beta$ is the effective coupling strength that depends on the parameters of the dynamical system as well as on external inputs (Supplementary Note 2.3). The reduced network of coupled binary On-Off units follows Glauber dynamics[34] (Supplementary Note 2.4), allowing us to calculate noise correlations analytically in our model. In simulations, both the dynamical-system and binary-unit versions of the network exhibit similar spatiotemporal dynamics, where the On and Off phases form local spatial clusters (Fig. 2d), which propagate across columns in a pattern of local irregular waves[29] (Supplementary Movie 1).

We model two sources of spiking variability: the On-Off fluctuations of the population activity and stochasticity of spike generation in individual neurons[21,35]. We simulate spikes of individual neurons as inhomogeneous Poisson processes with different mean rates during the On and Off phases generated by the network (Fig. 2e). This doubly stochastic description coincides with the assumptions of the HMM used to fit the experimental data. We match the model parameters to the experimental data by fitting the data with the HMM, which provides us with the estimates of the On-Off transition rates ($\alpha_1$ and $\alpha_2$) and the On and Off firing rates ($r_{on}$ and $r_{off}$) for all MUs and single units (SUs) in each recording session and task condition (Fig. 2f). We then use these parameters to predict noise correlations and compare these predictions with the values measured for the same neuron pairs in the data.

**The model accounts for correlated variability within columns.** We used the two-phase recordings to test how accurately the model predicts changes of correlated variability in single columns during attention. In the model, the On-Off dynamics are the source of correlations between responses of individual neurons. All neurons in the population represented by a single unit

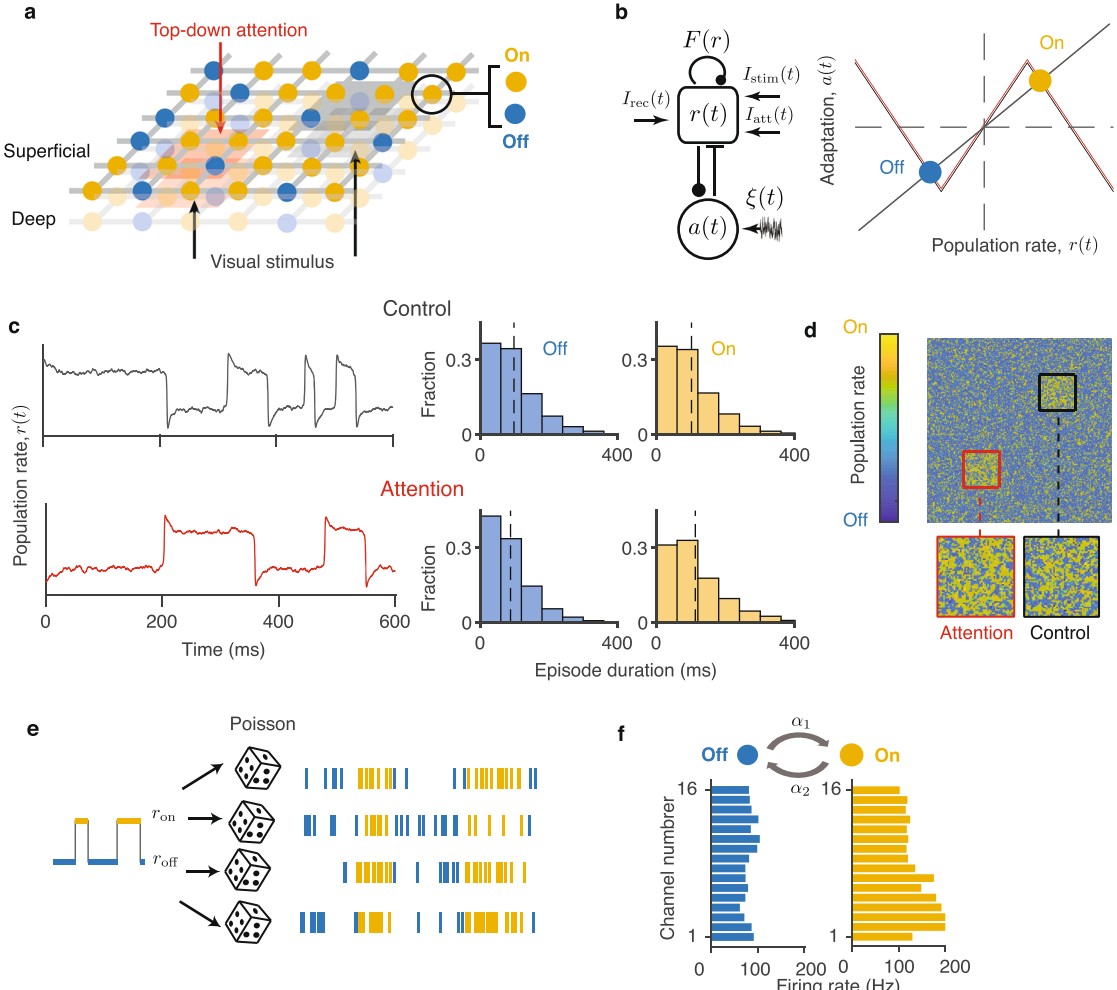

**Fig. 2 A network model of interacting cortical columns. a** The network consists of two parallel two-dimensional lattices corresponding to superficial and deep cortical layers. Each unit represents a local population of neurons within one layer of a cortical column. The units transition between the On (yellow) and Off (blue) phases. Bottom-up inputs from visual stimuli (gray shading) and top-down attentional inputs (red shading) target local groups of columns in the model. Each network layer receives different attentional inputs to account for differences in noise-correlation changes between superficial versus deep cortical layers. **b** Dynamical system modeling On-Off dynamics in single columns (left panel). The mean firing-rate variable $r(t)$ receives a recurrent self-coupling $F(r)$ and a negative feedback from the adaptation variable $a(t)$. The dynamical system is driven by a white noise $\xi(t)$, recurrent inputs from the neighboring columns $I_{rec}(t)$, and external inputs $I_{stim}(t)$ and $I_{att}(t)$. On the phase plane (right panel), the $r$-nullcline (gray) and $a$-nullcline (black) cross at the On (yellow) and Off (blue) stable fixed points. The attentional input shifts the $r$-nullcline (red) modulating the stability of the On and Off fixed points. **c** In single columns, the model generates stochastic On-Off transitions (left panel). The durations of On (yellow) and Off (blue) episodes are irregular and exponentially distributed (right panel). The average duration (dashed lines) of On-episodes is longer in attention ($I_{att} > 0$, lower row, average $\bar{\tau}_{on} = 113$ ms, $\bar{\tau}_{off} = 88$ ms) relative to the control condition ($I_{att} = 0$, upper row, average $\bar{\tau}_{on} = 102$ ms, $\bar{\tau}_{off} = 98$ ms). **d** The network generates spatiotemporal On-Off dynamics, where the On and Off phases form local spatial clusters (a single snapshot of simulated activity in the dynamical-system network is shown). The spatiotemporal pattern differs between attention (red square) and control (black square) conditions. **e** Spikes of individual neurons are modeled as inhomogeneous Poisson processes with different mean rates during the On (yellow) and Off (blue) phases generated by the network. All neurons represented by a single network unit follow the same shared On-Off sequence. **f** Model parameters are estimated by fitting the experimental data with the HMM, which provides the On-Off transition rates ($\alpha_1$ and $\alpha_2$, top) and the On and Off firing rates ($r_{on}$ and $r_{off}$) for each MU and SU in each recording session and task condition. Histograms show the On (yellow) and Off (blue) firing rates for MUs for an example HMM fit. Source data are provided as a Source Data file.

(column) follow the same sequence of On and Off episodes. The spiking responses differ across neurons because of the independent Poisson noises as well as differences in their On and Off firing rates. We derived analytical formulas for the Fano factor and noise correlations, measured over an arbitrary time-window $T$, as functions of the model parameters: the On-Off transition rates and the On and Off firing rates of each neuron (Methods section and Supplementary Note 2.1). The model analytically predicts the dependence of Fano factor and noise correlations on the measurement time-window[6], indicating that

this dependence is determined by the timescales of On-Off dynamics (Supplementary Note 2.1). We used these analytical formulas with the parameter values estimated from the data by the HMM to predict the FF and noise correlations for, respectively, each neuron and neuron pair in our dataset. We compared these model predictions with direct measurements from the experimental data.

Our model makes a specific prediction that the key factor determining the magnitude of FF and noise correlations within columns is the On-Off firing-rate difference $\Delta r = r_{on} - r_{off}$.

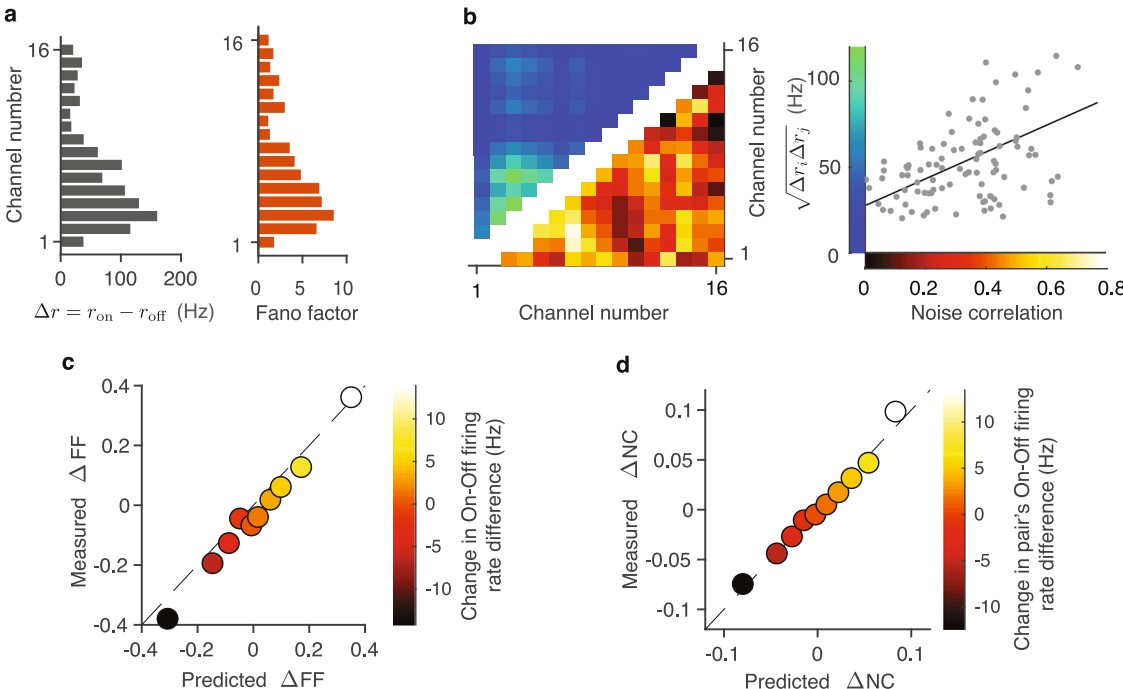

**Fig. 3 The model accounts for attentional modulation of correlated variability within columns. a** The model predicts that FF is proportional to the On-Off firing-rate difference $\Delta r = r_{on} - r_{off}$. For an example recording, $\Delta r$ ranges broadly across MUs (gray, left panel), and this variation of $\Delta r$ tightly corresponds with the variation of FF (orange, right panel). **b** The model predicts that noise correlation between neurons $i$ and $j$ is proportional to the product $\Delta r_i \Delta r_j$. For the same example recording, the variation in $\sqrt{\Delta r_i \Delta r_j}$ corresponds with the variation of noise correlations (left). Noise correlations are positively correlated with $\sqrt{\Delta r_i \Delta r_j}$, (right, black line - linear regression). **c** Comparison between attention-related changes ($\Delta FF = FF_{att} - FF_{ctl}$) in FF predicted by the On-Off dynamics model ($x$-axis) and measured directly from the data ($y$-axis) for MUs. All MUs are divided into 10 equally-sized groups based on the change in their On-Off firing-rate difference between attention and control conditions ($\Delta r_{att} - \Delta r_{ctl}$, color axis). **d** Comparison between attention-related changes ($\Delta NC = NC_{att} - NC_{ctl}$) in noise correlations (NC) predicted by the On-Off dynamics model ($x$-axis) and measured directly from the data ($y$-axis) for MUs. All MU-pairs are divided in 10 equally sized groups based on the change in the pair's On-Off firing-rate difference defined as $\sqrt{\Delta r_{att,i} \Delta r_{att,j}} - \sqrt{\Delta r_{ctl,i} \Delta r_{ctl,j}}$ (color axis). Source data are provided as a Source Data file.

Specifically, FF is directly proportional to $\Delta r$, and the noise correlation between neurons $i$ and $j$ is proportional to the product $\Delta r_i \Delta r_j$ (Methods section and Supplementary Note 2.1). This dependence on $\Delta r$ is intuitive because the source of correlations within a column is the shared On-Off switching, hence the stronger a neuron is modulated by the On-Off dynamics (the greater is $\Delta r$), the stronger it will be correlated with other neurons in the same column. The dependence of FF and noise correlations on $\Delta r$ is evident in an example recording (Fig. 3a, b), where different MUs exhibit a variety of On-Off firing-rate differences $\Delta r$. The FF ranges broadly across MUs, from ~1 up to ~9, and this variation is very well predicted by $\Delta r$ (Fig. 3a). Although a few units exhibited very high $\Delta r$ and hence unusually high FF values, the median FF of MUA was 1.8 similar to previous studies (Supplementary Fig. 6). Similarly, MU pairs with the largest product $\Delta r_i \Delta r_j$ also exhibit the largest noise correlations (Fig. 3b). While both FF and noise correlations also weakly depend on the On-Off transition rates, the On-Off firing-rate difference is the main factor defining the broad distributions of these quantities in single columns (Supplementary Fig. 7).

As a consequence, the model also predicts that the changes in FF and noise correlations during attention are proportional to the changes in $\Delta r$ and $\Delta r_i \Delta r_j$, respectively. This prediction was clearly borne out by the data: the measured change in FF had a strong trend as a function of the change in $\Delta r$ ($y$-axis vs. color-axis in Fig. 3c), and the measured change in noise correlations had a strong trend as a function of the change in $\Delta r_i \Delta r_j$ ($y$-axis vs. color-axis in Fig. 3d). Moreover, changes in the FF and noise correlations measured from the data were accurately matched by the model predictions ($y$- vs. $x$-axis in Fig. 3c, d). The attention-

related changes in noise correlations range widely across the population, with noise correlations substantially reduced and enhanced in many pairs. This entire broad distribution is accurately matched by the analytical predictions of the On-Off dynamics model in single columns. Despite substantial changes in many pairs, the average change in noise correlations within columns is near zero (Supplementary Fig. 8b), since the changes of $\Delta r_i \Delta r_j$ are broadly distributed but close to zero on average. Thus, our model of On-Off dynamics explains the observed changes in correlated variability during attention within columns.

**Decay of noise correlations with lateral distance.** Next, we analyzed the dependence of noise correlations on the lateral distance in our laminar recordings and in the network model. Previous studies in the visual cortex found that noise correlations decrease with the lateral distance[31,36,37]. These studies used multi-electrode arrays with lateral spacing between electrodes ranging from ~0.35 to 4 mm, i.e. sampling distant neuron pairs in different columns. With the laminar recordings, we tested how noise correlations depend on the lateral distance over a much shorter range of distances within single or nearby columns. We leveraged the fact that laminar recordings generally exhibit slight horizontal shifts due to variability in the penetration angle (Fig. 4a). As a surrogate for horizontal displacements between pairs of channels, we used distances between centers of their RFs. To estimate the range of physical distances in the cortex spanned by our laminar recordings, we converted the RF-center distances to cortical distances using the cortical magnification factor for each eccentricity[38]. The range of distances spanned by our

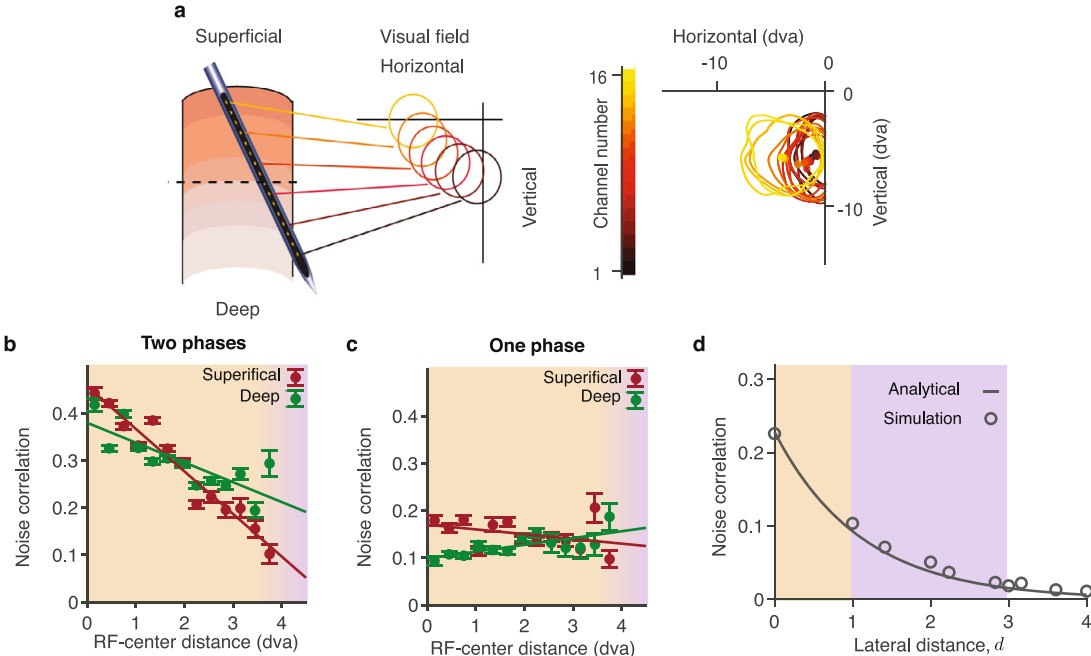

**Fig. 4 Dependence of noise correlations on lateral distance. a** Laminar recordings generally exhibit slight horizontal displacements which manifest in a systematic shift of the RFs (circles) across channels (left panel). The shift of the RFs (lines, RF contours; dots, RF centers; dva, degrees of visual angle) for an example recording (right panel). **b** In two-phase recordings, noise correlations decrease with the RF-center distance in both superficial (crimson) and deep (green) layers (dots - data points, lines - linear regression, superficial layers $n = 2544$ MU pairs; deep layers $n = 3064$ MU pairs). Orange background highlights the range of short lateral distances within single or nearby columns. Purple background highlights longer lateral distances between distant columns, such as distances covered by a Utah array, which are outside the range of our laminar recordings. Error bars represent the standard error of the mean (SEM). **c** Same as **b** for one-phase recordings. Noise correlations do not decrease with the RF-center distance (superficial layers $n = 920$ MU pairs; deep layers $n = 1448$ MU pairs). **d** Our theory predicts that noise correlations decay with lateral distance exponentially, with the decay constant $L$ called correlation length. Simulations of the full dynamical-system network (circles) agree with the analytical formula derived using the binary-unit network approximation (line). The model parameters $\alpha_1$, $\alpha_2$, $r_{on}$, and $r_{off}$ are sampled from a distribution of parameters in HMMs fitted to the data. Source data are provided as a Source Data file.

recordings was ~4−6 dva or ~1.5 mm (Methods section and Supplementary Fig. 9).

We found that noise correlations decreased with lateral distance in two-phase but not in one-phase recordings. In two-phase recordings, noise correlations monotonically decreased with the RF-center distance both in superficial and deep layers (Fig. 4b, linear regression, one-sided $t$-test, slope $-0.09 \pm 0.01$, $p < 10^{-8}$ in superficial layers, slope $-0.04 \pm 0.01$, $p < 10^{-3}$ in deep layers). With the conversion to cortical distances[38], noise correlations also decreased continuously with the lateral cortical distance over $\lesssim 1$ mm range (Supplementary Fig. 9). Note that most pairs in our data were at very short distances, with the median estimated cortical distance 0.72 mm. Thus the decay of noise correlations with lateral distance spans all distances within nearby and across distant columns. In contrast, noise correlations did not decrease with the RF-center distance in the one-phase recordings (Fig. 4c, linear regression, one-sided $t$-test, slope $-0.010 \pm 0.008$, $p = 0.1$ in superficial layers, slope $0.014 \pm 0.004$, $p = 0.997$ in deep layers). These results suggest that On-Off dynamics give rise to the lateral distance-dependence of noise correlations.

In our network model, the dependence of noise correlations on the lateral distance arises from the spatiotemporal On-Off dynamics. Whereas all neurons represented by a single unit in the network follow the same shared sequence of On-Off phases, neurons represented by different units follow their respective On-Off sequences. Consistent with this assumption, in recording sessions with large lateral shifts between receptive fields, we can

observe On-Off phases that occur synchronously only on a subset of adjacent channels and propagate across channels over time (Supplementary Fig. 10). The On-Off dynamics are correlated across nearby units in the lateral dimension due to horizontal connectivity in the network. Since the horizontal connections are spatially local and relatively weak, the synchrony of On-Off dynamics is not global across the entire network, but localized to a finite range of lateral distances. Thus the On-Off phases form spatial clusters with a characteristic spatial length scale, and beyond this spatial scale the On-Off phases are uncorrelated. This network mechanism leads to a continuous decrease of noise correlations with the lateral distance in the model (Fig. 4d).

Using the binary-unit reduced network model, we derived an analytical formula for the dependence of noise correlations on the lateral distance $d$ (Methods section). Our calculations show that noise correlations decay with the distance exponentially as $\mathcal{A} \exp(-d/L)$. This formula describes noise correlations both within and across columns. At zero distance ($d = 0$), the formula reduces to the pre-factor $\mathcal{A} = \mathcal{A}(\alpha_1, \alpha_2, \Delta r_i, \Delta r_j)$, which accounts for the dependence of noise correlations on the On-Off transition rates and the On-Off firing-rate difference, as described in the previous section. Across columns (finite $d > 0$), the formula accounts for the spatial structure of noise correlations with the exponential discount factor $\exp(-d/L)$. The space-constant $L$ of this exponential decay, termed correlation length, depends on the On-Off transition rates and on the effective coupling strengths $\beta$ between units in the network: $L = \sqrt{\beta/(\alpha_1 + \alpha_2)}$ (in dimensionless units of the lattice constant, see Methods section and

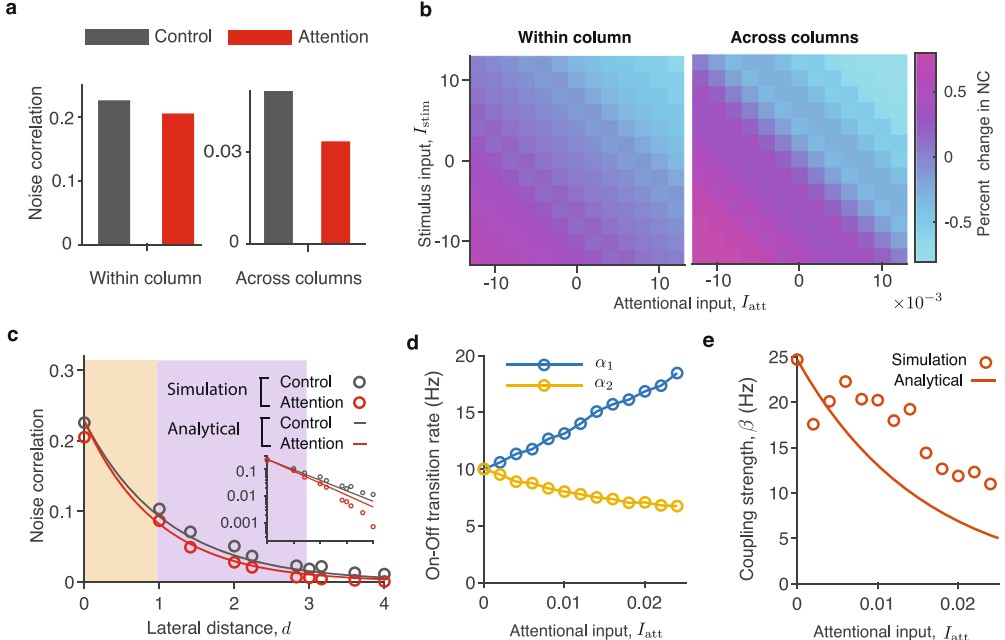

**Fig. 5 Attentional inputs modulate the efficacy of lateral interactions in the network leading to changes of the correlation length. a** In simulations of the dynamical-system network model, noise correlations between neurons in different columns robustly decrease during attention ($I_{att} > 0$, red) relative to control ($I_{att} = 0$, gray), while noise correlations between neurons within columns change only slightly. The average reduction of noise correlations is large across columns (MI$_{corr}$ = −0.21, right), but small within columns (MI$_{corr}$ = −0.05, left). **b** In simulations, noise correlations decrease with excitatory ($I_{att} > 0$) and increase with inhibitory ($I_{att} < 0$) attentional inputs. In all cases, the average changes of noise correlations within columns are small relative to sizable changes across columns. **c** Noise correlations decay faster with lateral distance in attention (red, $I_{att} > 0$) relative to control condition (gray, $I_{att} = 0$), hence the correlation length is reduced $L_{att} < L_{ctl}$. Data are shown from simulations of the full dynamical-system network (circles) and analytical calculations using the binary-network approximation (solid lines). Orange and purple backgrounds highlight the range of distances within and across columns, respectively. The change of the correlation length during attention is clearly visible in the same data plotted on the linear-logarithmic scale (inset). **d** The Off-to-On ($\alpha_1$, blue) and On-to-Off ($\alpha_2$, yellow) transition rates weakly depend on the attentional input in the dynamical-system network. **e** The effective coupling strength $\beta$ steeply decreases with increasing attentional input. Coupling strength $\beta$ estimated from simulations of the full dynamical-system network (circles) is compared to analytical calculations using the binary-network approximation (line, Supplementary Note 2.2).

Supplementary Note 2.3). This analytical result agrees well with simulations of the full dynamical-system network model (Fig. 4d).

The exponential decay of noise correlations with distance in our model, characterized by the correlation length $L$, is consistent with the decrease in noise correlations observed over a wide range of cortical distances spanned by our laminar (Fig. 4b) and previous lateral recordings[31,36,37] from the primate visual cortex. Our model can also reconcile the decay of noise correlations with distance in lateral recordings[31,36,37] with the lack of distance dependence in the one-phase recordings. With some heterogeneity, if a random fraction of units in the model does not exhibit On-Off transitions (due to a more stable fixed point), the activity of these one-phase units is not correlated with other units at all distances. Thus lateral sampling from a mixture of one-phase and two-phase phase units would uniformly lower the average of noise correlations without affecting their distance dependence.

**Differential changes in noise correlations arise from attentional modulation of the correlation length.** In our network model, attentional inputs restructure the spatiotemporal On-Off dynamics, leading to differential changes in noise correlations within versus across columns. In the network simulations, a local group of units receives an attentional input $I_{att}$, while other units without this input ($I_{att} = 0$) are in the unattended control condition (Fig. 2d). With an excitatory attentional input ($I_{att} > 0$), average noise correlations between neurons in the same column change very little relative to control (MI$_{corr}$ = −0.05), while noise correlations between neurons in different columns are substan-

tially reduced (MI$_{corr}$ = −0.21, Fig. 5a). These results replicate the order-of-magnitude difference of MI$_{corr}$ observed between laminar versus lateral recordings from the visual cortex. We repeated simulations for a range of excitatory ($I_{att} > 0$) and inhibitory ($I_{att} < 0$) attentional inputs. The excitatory inputs reduced noise correlations, whereas inhibitory inputs increased noise correlations, but in all cases, the average changes of noise correlations within columns were smaller compared to sizable changes across columns of the network (Fig. 5b).

To reveal the mechanism leading to differential changes of noise-correlations within versus across columns, we examined how attentional inputs affect the dependence of noise correlations on the lateral distance in our network. In simulations, excitatory attentional inputs produce a faster decay of noise correlations with lateral distance, which corresponds to a shorter correlation length ($L_{att} < L_{ctl}$, Fig. 5c and Supplementary Fig. 11). Due to this faster spatial decay, noise correlations at intermediate lateral distances (finite $d > 0$) are considerably lower in attention relative to control conditions, even when changes of noise-correlations within columns ($d = 0$) are small. Inhibitory attentional inputs, on the other hand, produce a slower decay of noise correlations with lateral distance, which corresponds to a longer correlation length ($L_{att} > L_{ctl}$) and results in increase of noise correlations at intermediate lateral distances. Thus changes of the correlation length $L$ produce sizable changes of noise correlations at intermediate lateral distances (across columns) even when noise correlations within columns do not change.

To understand the network mechanism by which attentional inputs modulate the correlation length, we leveraged the analytical

formula $L = \sqrt{\beta/(\alpha_1 + \alpha_2)}$. In the dynamical-system model, an excitatory attentional input shifts the $r$-nullcline, which increases Off-to-On ($\alpha_1$) and decreases On-to-Off ($\alpha_2$) transition rates (Fig. 2b, c). Since $\alpha_1$ and $\alpha_2$ change only moderately and in opposite directions, their sum remains nearly constant (Fig. 5d). Therefore changes of the correlation length $L$ are mainly driven by changes in the effective coupling strength $\beta$, which decreases steeply with an increasing attentional input (Fig. 5e). The effective coupling strength $\beta$ decreases because an excitatory attentional input stabilizes the On fixed point, thereby effectively reducing the efficacy of the lateral recurrent inputs to drive the On-Off transitions. Vice versa, an inhibitory attentional input makes the On fixed point less stable, thereby enhancing the relative efficacy of the lateral recurrent inputs and hence extending the spatial correlation length in the network (Supplementary Note 3.2). Thus the attentional input modulates the correlation length by regulating the relative efficacy of lateral interactions between columns[39], which leads to differential changes in noise correlations within versus across columns.

**The model predicts distance-dependent changes of noise correlations**. The major changes in noise correlations in our model are driven by changes in the correlation length $L$. The model, therefore, makes a specific prediction that changes in noise correlations during attention are not uniform across space. Noise correlations decay exponentially with the lateral distance, with different decay rates in attention and control conditions. Hence the spatial profile of noise-correlation changes is defined by the difference of two exponential decays: $\exp(-d/L_{\mathrm{att}})$ and $\exp(-d/L_{\mathrm{ctl}})$. At very short lateral distances (within columns), average changes of noise-correlations are small (Fig. 1d). At very long lateral distances, the average changes are negligible, because the overall magnitude of noise correlations vanishes. Sizable changes in noise correlations are predicted to occur at intermediate lateral distances, where the difference between two exponential decays dominates. Thus the network mechanism in our model predicts that the magnitude of noise-correlation changes depends on lateral distance (Fig. 6a, b). This prediction contrasts with the alternative balanced network model, where population activity fluctuates around a single global fixed point, which instead predicts that attention-related changes of noise correlations are spatially uniform[31] (Fig. 6c and Supplementary Note 3). Therefore examining the spatial profile of noise-correlation changes in the data could distinguish between network mechanisms these alternative models postulate.

We analyzed how changes in noise correlations during attention depend on the lateral distance (estimated by the RF-center distance) in our laminar recordings. Although average noise-correlation changes in our columnar recordings were small, when sorted by the lateral distance, the data revealed spatial patterns with substantial changes in noise correlations at longer distances. As predicted by our model, changes of noise correlations in the two-phase recordings were not uniform across space. In the superficial layers, changes of noise correlations were smallest at very short lateral distances and became progressively larger at longer distances (Fig. 6d). Noise correlations decreased during attention, with greater reduction at longer distances (linear regression, one-sided $t$-test, slope $-0.017 \pm 0.004$, $p < 10^{-3}$). The extrapolation of this trend to intermediate lateral distances ($1 \leqslant d \leqslant 4$ mm) is consistent with a robust reduction of noise correlations during attention observed in Utah-array recordings, which also sample from superficial layers[10,12]. In the deep layers, the attentional effects on noise correlations were reversed from that in superficial layers, showing a moderate increasing trend with a borderline statistical effect (Fig. 6e, linear regression, one-sided $t$-test, slope

$0.006 \pm 0.004$, $p = 0.06$; $t$-test for slopes superficial versus deep: $p = 0.6 \times 10^{-3}$). In contrast, changes of noise correlations did not depend on lateral distance in the one-phase recordings (Fig. 6f, linear regression, $t$-test, slope $-0.0005 \pm 0.0061$, $p = 0.93$ in superficial layers and slope $-0.01 \pm 0.01$, $p = 0.35$ in deep layers). These results indicate that On-Off dynamics give rise to distance-dependence of noise-correlation changes during attention.

The spatial profile of noise-correlation changes in two-phase recordings is consistent with our network mechanism but inconsistent with the previously proposed model, which predicts spatially uniform changes[31]. The observed spatial profiles of noise-correlation changes indicate that the correlation length decreases in superficial but not in deep layers, which suggests that superficial and deep layers receive different modulatory inputs during attention[40].

### Discussion

Our results show that On-Off dynamics are a major source of correlated variability in the visual cortex. On-Off dynamics in the awake cortex resemble some features of Up-Down transitions prominent during slow-wave sleep and anesthesia. Up and Down states were originally used to describe the bimodal distribution of membrane potentials, and now are also used to refer to spiking (Up) and silent (Down) phases of population activity during slow-wave sleep and anesthesia[5]. Up-Down transitions are a major source of noise correlations during anesthesia[21,41]. Similarly, On-Off dynamics account for a dominant share of noise correlations in behaving animals.

We found that On-Off dynamics explained the magnitude of noise correlations, their attentional modulation and dependence on lateral distance. The average changes of noise correlations during attention were very small within columns of the visual area V4. Noise correlations slightly decreased in superficial and increased in deep layers, but the changes were an order of magnitude smaller than a robust and sizable reduction of noise correlations between neurons in different columns. A reduction of noise correlations was suggested to be a major contributor to the improved behavioral performance during spatial attention[10]. Our results show, however, that changes of noise correlations are not uniform: their magnitude and sign depend on the relative anatomical positions of neurons within layers and columns of the visual cortex. These heterogeneous changes of noise correlations may reflect unique contributions to behavioral improvements from different functional groups of neurons defined by their anatomical positions within the circuit.

To explain differences in attention-related changes of noise correlations within versus across columns, we developed a network model of interacting cortical columns. The key mechanism generating correlated variability in the model is On-Off dynamics, metastable transitions between the high and low firing-rate fixed points in single columns. On-Off dynamics propagate laterally across columns via spatially structured network connectivity to form activity clusters traveling as local irregular waves (Supplementary Movie 1). Our model, therefore, integrates experimental findings that spontaneous fluctuations of cortical activity follow bistable On-Off dynamics in single columns[22] and propagate laterally across columns as waves[30]. Due to the stochasticity of dynamics, the activity clusters do not propagate coherently across the entire network but travel only locally until they fade or merge with other clusters. Local irregular waves differ from global traveling waves, in which a wave can propagate coherently across the entire network and most neurons equally participate in each wave[29]. The spatial scale of activity clusters defines the exponential decay constant of noise correlations with lateral distance, i.e. the correlation length. Attentional inputs restructure the

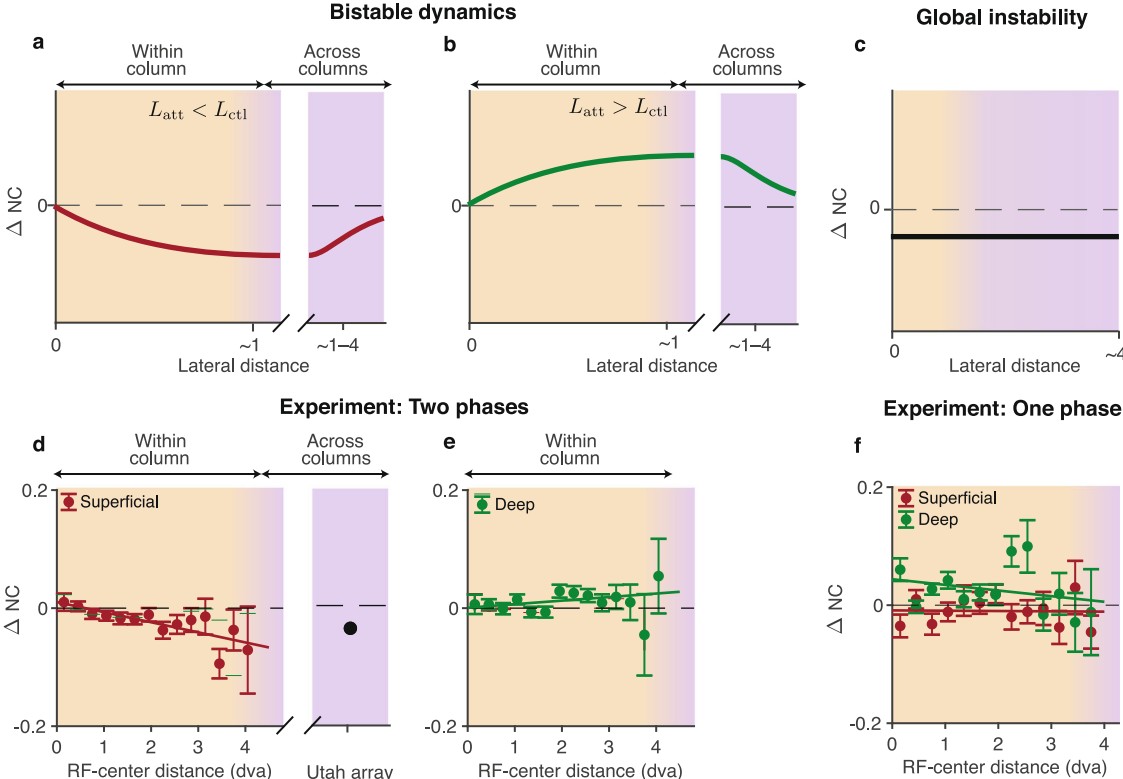

**Fig. 6 Attentional changes in noise correlations depend on lateral distance. a** Our model predicts that changes in noise correlations at intermediate lateral distances are driven by changes of the correlation length $L$. When the correlation length decreases ($L_{att} < L_{ctl}$), the model predicts a robust reduction of noise correlations at intermediate distances (across columns, purple background) when changes at zero distance are vanishing (crimson line). Over short distances (within the column, orange background), noise correlations progressively decrease with distance. **b** When the correlation length increases ($L_{att} > L_{ctl}$), our model predicts an increase of noise correlations at intermediate distances when changes at zero distance are vanishing (green line). In both cases (**a**, **b**), the spatial profile of noise-correlation changes is not uniform, and sizable changes are predicted to occur at intermediate lateral distances. **c** The balanced network model with fluctuations around a single global fixed point predicts uniform changes in noise correlations at all distances (black line)[31]. **d** In two-phase recordings, changes in noise correlations during attention depend on the RF-center distance. The magnitude of noise-correlation changes is vanishingly small at zero distance and progressively increases at longer distances. With increasing RF-center distance, noise correlations decrease in superficial layers (crimson). Data for single columns (crimson dots - data, line - linear regression, $n = 2544$ MU pairs) are shown along with an approximate value of changes in noise correlation between neurons in different columns recorded with a Utah array (black dot). Error bars represent SEM. **e** Same as **d** for deep layers in two-phase recordings ($n = 3064$ MU pairs). With increasing RF-center distance, noise correlations show a moderate increasing trend with borderline statistical effect. **f** Same as **d**, **e** for one-phase recordings. Changes in noise correlations during attention do not depend on the RF-center distance in superficial (crimson) and deep (green) layers (superficial layers $n = 920$ MU pairs; deep layers $n = 1448$ MU pairs). Source data are provided as a Source Data file.

spatiotemporal On-Off dynamics and modulate the correlation length, which results in distance-dependent changes of noise correlations. The model qualitatively captures attention-related changes of noise correlations in our data. Moreover, it makes a testable prediction that the sizable changes of noise correlations occur at intermediate lateral distances. Consistent with this prediction, we found that in our laminar recordings, the magnitude of noise-correlation changes gradually increased with lateral distance in superficial layers (a moderate trend with borderline statistical effect in deep layers). These results show that changes in noise correlations depend on lateral distance.

On-Off dynamics accounted for the dominant share of noise correlations in the majority of our recordings (two-phase recordings), while in some recordings (one-phase recordings) HMM did not detect On-Off transitions. Although noise correlations in one-phase recordings were substantially smaller than in two-phase recordings, they were nonzero, indicating that other sources of variability contribute to noise correlations besides On-Off dynamics. Since in one-phase recordings, noise correlations did not depend on distance (Fig. 4c), these other variability sources may be more global and uniform within a cortical area,

such as fluctuations in neural excitability related to arousal[2,5,22]. Moreover, Fano factor and noise correlations in one-phase recordings were modulated during attention, suggesting that the additional variability sources also interact with attentional mechanisms, producing spatially uniform changes in correlated variability (Fig. 6f).

Noise correlations can limit stimulus information encoded in the population, meaning that information saturates with increasing population size[15,16,18,19]. Information saturation is caused by a specific pattern of correlations, known as differential correlations, which are proportional to the product of the derivatives of the tuning curves[17,42]. In our model, assuming that stimulus does not change the statistics of On-Off dynamics and that changes in the On-Off dynamics do not affect stimulus tuning, we found that the On-Off dynamics influence the strength of differential correlations and thus affect the saturation level of information (Supplementary Note 4). Specifically, the linear Fisher information is monotonically decreasing with the correlation length. Hence, a reduction of the correlation length leads to an increase in stimulus information, as we observed in superficial cortical layers during attention. However, the On-Off dynamics

and stimulus tuning are likely interdependent in the cortical circuitry, where both arise from the same structured connectivity. Deeper understanding of how On-Off fluctuations impact sensory coding will be possible using models with connectivity that supports stimulus tuning in addition to spatial receptive fields[43] in future work.

Several mechanisms were proposed to explain how correlated fluctuations arise in cortical networks. There are two general classes of models: one relies on external shared variability and another generates shared variability via intrinsic network dynamics. In models with external shared variability, the source of correlated fluctuations is assumed to be outside the network, and the network merely filters the correlated input noise[32,44,45]. In most of these models, the mechanism is based on a spatial connectivity structure that locally breaks the excitation-inhibition balance. The classical balanced network model with random connectivity[46,47] operates in an asynchronous regime, where the tight excitation-inhibition balance cancels any input correlations resulting in zero average noise correlations[48]. The spatial connectivity structure, where recurrent inhibition is broader than feedforward excitation, breaks the balance locally, hence the input correlations cannot be canceled resulting in positive average noise correlations[44,45]. However, to match the experimentally observed temporal and spatial scales of correlations, all of these models have to assume ad hoc spatiotemporal structure of the input noise[32,44,45].

The second class of models can generate shared variability internally. One mechanism is based on breaking stability in some spatial Fourier modes in a spatially organized balanced network. For example, in a two-dimensional balanced network with slow inhibitory kinetics, shared fluctuations arise from instability at some spatial frequency that generates rate chaos[31]. Similarly, in a one-dimensional balanced ring model, strong correlations arise from a feed-forward structure in some Fourier modes of connectivity[49]. In these models, correlations arise from fluctuations around a global fixed point with a timescale defined by the mismatch between excitatory and inhibitory synaptic time-constants, i.e. just a few milliseconds. This fast timescale is inconsistent with experimental data as fluctuations of cortical activity occur on a timescale of about a hundred milliseconds[50] and exhibit signatures of metastable dynamics[22–24].

An alternative mechanism that can account for the slow timescale of cortical fluctuations is based on multi-stability. In this case, slow correlated fluctuations arise from stochastic transitions between multiple fixed points in the network. Multi-stability can result from clustered excitatory connectivity, where each cluster corresponds to a fixed-point attractor[51]. Further, bistability between high and low firing-rate attractors can arise in unstructured networks with strong recurrent excitation and slower negative feedback such as firing-rate adaptation[21,26,52,53] or short-term synaptic depression[54]. Our model of On-Off dynamics is based on bistability, which is consistent with exponential distributions of On and Off episode durations in behaving monkeys[22]. Similar models with slow negative feedback were used to reproduce Up-Down dynamics[26,55]. This mechanism can generate slow alternations between high and low firing rates via several dynamical regimes. In particular, Up-Down dynamics were found to be bistable during anesthesia[26] and excitable during slow-wave sleep[55] (a single stable fixed point from which suprathreshold fluctuations induce large transient events). The models with multi-stability capture the slow timescale of cortical fluctuations and produce realistic noise correlations[21,26,53]. However, the multi-stable networks studied previously were not endowed with a spatial connectivity layout akin to organization of cortical networks, hence they do not produce any spatial structure of noise correlations.

To account for both the slow timescale and spatial structure of noise correlations in the visual cortex, our network model combines local bistable On-Off dynamics with spatially organized connectivity. In our model, correlated variability arises from metastable transitions between the On and Off fixed points, and not from fluctuations around a single global attractor[32]. Our results suggest that a theory of noise correlations in the visual cortex should incorporate both the anatomical connectivity structure of visual areas as well as the local bistability of population dynamics in single columns.

Recurrent network models were previously developed to suggest possible circuit mechanisms that produce a reduction of noise correlations during attention[31,32,56]. These models are based on a dynamical mechanism, where the network operates around a global fixed point and attentional inputs increase the stability of this fixed point leading to suppression of correlated fluctuations. Specifically, in the network with intrinsically generated shared variability, the stability of the operating point can be increased by up-regulating activities of inhibitory neurons[31]. However, elevated inhibition reduces firing rates of excitatory neurons, which contradicts attentional enhancement of firing rates in experiments. In the network filtering external noise, the stability of the global fixed point can be increased by excitatory inputs when the network operates in inhibition dominated regime[32]. In this scenario, an excitatory input increases effective lateral connectivity, which suppresses the transmission of the correlated input noise (Supplementary Note 3.4).

The mechanism we propose differs from these previous models. First, we show that a reduction of noise correlations during attention is not universal. Therefore, a network mechanism should account for heterogeneous changes of noise correlations across different anatomical dimensions. Second, the mechanism in our model is based on the local bistability of On-Off dynamics in single columns. Attentional inputs change the stability of the On and Off fixed points, which effectively modulates the efficacy of lateral interactions across the network leading to changes of the correlation length. This mechanism is fundamentally distance-dependent, as the major changes of noise correlations in our model are driven by changes of the correlation length (Supplementary Note 2.3, 3.2). As a consequence, we find that in Fourier space the lower spatial frequency modes contribute most to noise-correlation changes (Supplementary Note 3.2). This result partially agrees with the previous model[31], where the dominant part of noise-correlation changes arises from zero spatial frequency mode, which, however, predicts a spatially uniform modulation of noise correlations. In contrast, contributions from higher spatial frequency modes are not negligible in our model. A combination of all spatial frequency modes generates a non-monotonic profile of noise-correlation changes in lateral dimension, a prediction that was confirmed in our data.

Several biophysical substrates could mediate the network mechanism of attentional modulation in our model. Top-down projections from frontal cortical areas, especially Frontal Eye Fields (FEF)[57,58] can provide temporally and spatially precise inputs to drive fast and local modulation of On-Off dynamics in the visual cortex. Most FEF projections to V4 target pyramidal neurons[40], in agreement with our model where reduction of noise correlations in superficial layers is driven by external excitatory inputs, and unlike models where reduction of noise correlations is mediated by inputs to inhibitory neurons[31]. Neuromodulatory inputs can also mediate effects of attention[59] and can influence On-Off dynamics by modulating neural excitability and firing-rate adaptation[60]. The level of Acetylcholine (ACh) can modify the efficacy of synaptic interactions during attention in a selective manner[61]. An increase in ACh strengthens the thalamocortical synaptic efficacy by affecting nicotinic receptors and reduces the

efficacy of horizontal recurrent interactions by affecting muscarinic receptors. A decrease in the efficacy of horizontal interactions leads to a reduction of correlation length in our model. Further, laminar distribution of top-down inputs[40] and of neuromodulation, combined with layer-specific horizontal connectivity could account for the differential modulation of noise correlations in superficial and deep layers that we observed. Identifying precise mechanisms by which these multiple biophysical components interact within a columnar microcircuit is an important direction for future work.

## Methods
The research complies with all relevant ethical regulations. Experimental procedures were in accordance with NIH Guide for the Care and Use of Laboratory Animals, the Society for Neuroscience Guidelines and Policies, and Stanford University Animal Care and Use Committee.

**Behavior and electrophysiology**. Two male monkeys (*Macaca mulatta*, 8−12 kg, between 6 and 9 years of age) were used in experiments. The monkeys were trained on a cued change-detection task[22,62]. The monkey was required to make a difficult visual discrimination at a peripheral location with a central cue indicating which location would contain the change. The monkey was rewarded for reporting a successful detection with a saccade to the diametrically opposite peripheral location (antisaccade response). On each trial, a small central cue indicated the stimulus that was most likely to change. The cued stimulus was therefore a target of covert attention. The monkeys reported stimulus changes with an antisaccade to the location opposite to the change, which was therefore a target of overt attention due to anticipation of antisaccadic response[22,62] (Supplementary Fig. 1). Modulations of neural responses in V4 were highly similar during the covert and overt attention, including changes in firing rates, spiking variability and noise correlations[22,62], and therefore we combined the covert and overt attention conditions in our analyses. The monkey initiated each trial by fixating a central fixation dot on the screen. Within several hundred milliseconds, four peripheral stimuli appeared (static Gabor patches: oriented black and white gratings in a circular Gaussian aperture). After a short delay, the attention cue appeared: a short line originating at the fixation dot and extending in the direction of one of the four stimuli, randomly chosen on each trial with equal probability. The cue indicated with ~90% validity which of the four stimuli, if any, would change on each trial. After a post-cue period of 600−2300 ms, all stimuli synchronously disappeared for a brief interval and then reappeared. On half of the trials, one of the four stimuli reappeared with a changed orientation (i.e. rotated in place), and the monkey was rewarded for performing a saccadic eye movement to the location opposite to the changed stimulus. On the other half of the trials, all stimuli reappeared with the same orientations as they had before disappearing, and the monkey was rewarded for maintaining fixation on the central dot.

While monkeys performed the attention task, recordings were made in the visual area V4 with a 16-channel linear array microelectrodes[22,62]. The total length of array is 2.25 mm, and the spacing between electrical contacts is 150 μm. Recordings were targeted with MRI to be as perpendicular to cortical layers as possible so as to maximize the overlap of receptive fields (RFs) of recorded neurons. Each of the recording channels was assigned laminar depth relative to a common current source density marker as described previously[33].

**Data analysis**. Data were analyzed with a custom code written in Matlab. We measured Fano factor and noise correlations in our recordings using spike-counts $N$ of MUA and SUA in 200 ms bins (400–600 ms window after the attention cue onset). The Fano factor is the ratio of the spike-count variance to its mean across trials: $\mathrm{Var}[N]/\mathrm{E}[N]$. The noise correlation $r_{sc}$ is the Pearson correlation coefficient between spike-counts $N_i$ and $N_j$ of two neurons:

$$r_{sc} = \frac{\mathrm{E}[N_i N_j] - \mathrm{E}[N_i]\,\mathrm{E}[N_j]}{\sqrt{\mathrm{Var}[N_i]\,\mathrm{Var}[N_j]}}. \quad (1)$$

We estimated parameters of the On-Off dynamics in single columns by fitting population spiking activity in our recordings with a two-state Hidden Markov Model (HMM) as described previously[22]. HMM has a latent variable representing an unobserved population state that stochastically switches between the On and Off phases following Markov dynamics. Spikes on 16 simultaneously recorded channels are assumed to be generated by inhomogeneous Poisson processes, with different mean rates during the On and Off phases. The latent On-Off state is shared by the population, but the On and Off firing rates can differ across neurons. HMM was fitted to MUA spike-counts in 10 ms bins, during a time-window starting at 400 ms after the attention cue onset and until the end of the post-cue period. The duration of this time-window ranges between 200 and 1900 ms across trials. HMM was fitted separately for each of the 32 task conditions (4 attention conditions × 8 grating orientations). The HMM parameters were optimized with the Expectation-Maximization algorithm[22]. The HMM had 34 parameters: firing

rates in the On ($r_{on}$) and Off ($r_{off}$) phases for each of 16 channels and transition probabilities $p_{on}$ and $p_{off}$ for the entire population.

We selected the optimal number of HMM states using the same cross-validation procedure as in our previous work[22]. We computed average cross-validation error for HMMs with $n$ phases ($n = 1, \ldots 8$), normalized by the average cross-validation error of the HMM with 1 phase. For each recording session, the 4-fold cross-validation error was computed in 200 ms windows for each condition (4 attention conditions × 8 stimulus orientations), and then averaged across all channels, conditions and cross-validation folds. For most recordings, addition of the second phase greatly reduced cross-validation error compared to the single-phase HMM, whereas adding more phases resulted in only marginal improvements: the error curves display an elbow at $n = 2$, suggesting that 2-phase HMM is the most parsimonious model for our data. For some recording sessions, HMMs with $n > 1$ phases did not perform better than the 1-phase HMM and the error curves did not exhibit a kink for these recordings. We defined these to be one-phase recordings.

We estimated lateral shifts between channels in our laminar recordings by distances between centers of their RFs. The RF mapping procedure was described previously[22]. RFs were measured by recording spiking responses to briefly flashed stimuli on an evenly spaced 6 × 6 grid covering the lower left visual field. Spikes in the window 0–200 ms relative to stimulus onset were averaged across all presentations of each stimulus. The RF center was defined as the center of mass of the response map. The lateral cortical distance $d_{cortical}$ (mm) was estimated from the RF-center distance $d_{RF}$ (d.v.a) using the cortical magnification factor $M$ for each eccentricity $E$[38]:

$$d_{cortical}\text{(mm)} = 9 - M \times d_{RF}\text{ (d.v.a)}, \quad M = 3.1E^{-0.9}. \quad (2)$$

**Network model of interacting columns**. The model describes spatiotemporal dynamics of neural population activity across the cortical surface. The network consists of two two-dimensional square lattices of units, representing superficial and deep cortical layers. Each unit represents a local population of neurons within one layer of a single column. The dynamical variable $r(\mathbf{x}, t)$ represents the mean firing rate of this population. The two-dimensional lateral coordinates are denoted as $\mathbf{x}$. The dynamics of the network model are given by

$$\epsilon \frac{d}{dt} r = F(r) - a + W\nabla^2 r + I_{stim} + I_{attn},$$
$$\frac{d}{dt} a = gr - a + f + \sqrt{2Q}\xi. \quad (3)$$

Here $a(\mathbf{x}, t)$ is the adaptation variable, $\xi$ is a white Gaussian noise of unit intensity, and we omit the spatial indices of variables $r$ and $a$ for clarity. $\epsilon \ll 1$ is a constant that separates the timescales of the fast dynamical variable $r$, and slow adaption variable $a$ (Supplementary Note 2.2.3 and Supplementary Fig. 12).

The noise term $\xi$ in the adaptation variable drives stochastic transitions between the On and Off phases in single columns. This phenomenological noise term models fluctuations in population activity due to internal spiking noise and/or external stochastic inputs. Spiking noise can arise from finite-size fluctuations[63] as well as from biophysical sources, such as stochasticity of ion channels or synaptic release. External inputs correlated on the spatial scale of a column could also probabilistically drive On-Off transitions, for example, inputs related to microsaccades[22]. The term $\xi(t)$ models the net effect of various biological noise sources. Including the noise in the equation for adaptation variable enables analytical reduction of the dynamical system to the binary-unit model[64] (Supplementary Note 2.2.4). Adding the noise term in the firing-rate variable produces qualitatively similar results (Supplementary Fig. 13).

The function $F(r)$ is given by

$$F(r) = \begin{cases} -1 - r, & r \leq -1/2 \\ r, & -1/2 < r < 1/2 \\ 1 - r, & r \geq 1/2. \end{cases} \quad (4)$$

This piecewise linear function approximates the inverted N-shaped $r$-nullcline, typically used in rate-models with adaptation[21], which allows us to analytically reduce the dynamical system to a binary-unit model[64]. The term $W\nabla^2 r$ represents lateral interactions between neighboring units, where $\nabla^2 r = \partial_x^2 r + \partial_y^2 r$ implements a diffusive coupling and $W$ is the interaction strength parameter. The external currents $I_{stim}$ and $I_{attn}$ are applied to local groups of units to model stimulus and attentional inputs, respectively. A constant $\epsilon \ll 1$ separates the timescales of the fast firing-rate variable $r$ and slow adaptation variable $a$. The parameters $g, f, Q$ are chosen so that the system is bi-stable[64], where the population rate $r$ stochastically switches between two stable fixed points, corresponding to the On and Off phases.

We match the model to experimental data using the fitted HMM parameters. Specifically, the HMM transition matrix $P$ ($p_{11} = p_{off}$, $p_{12} = 1 - p_{off}$, $p_{22} = p_{on}$, $p_{21} = 1 - p_{on}$) provides an estimate of the On-Off transition rates: $\alpha_1 = (1 - p_{off})/\Delta t$ and $\alpha_2 = (1 - p_{on})/\Delta t$, where $\Delta t = 10$ ms is the bin size used for HMM fitting. HMM also estimates the On and Off firing rates $r_{on}$ and $r_{off}$ for each MUA and SUA, which we use to generate spikes of the model neurons. To this end, for each network unit we segment the simulated time-series $r(t)$ into the On ($S = 1$) and Off ($S = 0$) phases as $S(t) = \Theta[r(t)]$, where $\Theta$ is the Heaviside step function. The spike counts are then generated from inhomogeneous Poisson processes with rates $r_i(t)$,

where the firing rate for neuron $i$ is

$$r_i(t) = r_{\text{off},i} + \Delta r_i S_i(t), \quad \Delta r_i = r_{\text{on},i} - r_{\text{off},i}. \tag{5}$$

**Simulations.** We simulated the network model Eq. (3) on a $256 \times 256$ discrete square lattice with a time step of 0.005 s. The unit activities are initialized randomly. We compute noise correlations from 100 simulated trials for each set of parameters. On each trial, we simulated the period of spontaneous activity, stimulus period and attention-cue period, as in the experimental data. During stimulus period, external inputs $I_{\text{stim}}$ were applied to two local groups of units with the size $50 \times 50$. During the attention-cue period, one of these two groups also received attentional inputs $I_{\text{att}}$. To calculate noise correlations, we either assigned fixed values of $r_{\text{on}}$ and $r_{\text{off}}$ or sampled them from distributions of $r_{\text{on}}$ and $r_{\text{off}}$ extracted from experimental data by HMM.

**Reduction to a binary-unit network.** When the dynamical-system network operates in the bistable regime, the activity of each unit $i$ can be approximated by a binary variable $S_i$,[64] where $S_i = 1$ refers to On phase, and $S_i = 0$ to Off phase. We derived a reduced network model, where the dynamical equations describe the state transition probabilities of binary units. Using the mean-field approximation, we derived an approximate form for transition rates of binary units (Supplementary Note 2.3). In the leading approximation order, we have

$$w(S_i = 0 \to 1) \approx \alpha_1 + \beta_1 S_{i\pm1}, \ w(S_i = 1 \to 0) \approx \alpha_2 - \beta_2 S_{i\pm1}. \tag{6}$$

Here $S_{i\pm1}$ are the sum of activities of neighboring units that are connected to a given unit $S_i$. $\alpha_1, \alpha_2,$ and $\beta_1, \beta_2$ are functions of parameters in the dynamical-system model: $f, g, Q, I_{\text{stim}},$ and $I_{\text{attn}}$. This reduced model allows us to derive analytical formulas for correlations between units in the network.

**The reduced network model of binary units.** The binary-unit network operates on a two-dimensional square lattice. The network consists of $N$ units. Each unit can be in a discrete On ($S_i = 1$) or Off ($S_i = 0$) state, represented by a binary variable $S_i = \{0, 1\}$, ($i = 1, \ldots, N$). At time $t$, the probability of the system to be in a certain configuration $\{S\} = \{S_1, S_2, \ldots, S_N\}$ is denoted as $P(\{S\}, t)$. The rate of change of $P(\{S\}, t)$ is described by the master equation:

$$\frac{d}{dt} P(\{S\}, t) = -P(\{S\}, t) \sum_i w(S_i) + \sum_i P(\{S\}^{i*}, t) w(1 - S_i). \tag{7}$$

Here $\{S\}^{i*} = \{S_1, S_2, \ldots, 1 - S_i, \ldots, S_N\}$, and $w(S_i)$ is the transition rate. When $S_i = 0$, the transition rate of $S_i$ from 0 to 1 is

$$w(S_i = 0) = \alpha_1 + \beta_1(S_{i\pm1}). \tag{8}$$

When $S_i = 1$, the transition rate of $S_i$ from 1 to 0 is

$$w(S_i = 1) = \alpha_2 - \beta_2(S_{i\pm1}). \tag{9}$$

Here $\alpha_1$ and $\alpha_2$ represent the baseline transition rates of each unit without interactions with other units, and $\beta_{1,2}$ describe how the transition rates are influenced by nearby units $S_{i\pm1}$. The diffusive coupling between units is described by the discrete Laplacian:

$$S_{i\pm1} = S_{i+1} - S_i + S_{i-1} - S_i. \tag{10}$$

For simplicity, we use a single index $i$ to represent indices in arbitrary dimension. For example, in two dimensions $i = (x, y)$, and we have

$$S_{i\pm1} = S_{x+1,y} - S_{x,y} + S_{x-1,y} - S_{x,y} + S_{x,y+1} - S_{x,y} + S_{x,y-1} - S_{x,y}. \tag{11}$$

The dynamics of the binary-unit network resemble Glauber dynamics of the 2-D Ising model. However, in general, the detailed balance condition does not hold in the binary-unit model, so its dynamics are different from the 2-D Ising model (Supplementary Note 2.4).

Based on the master equation, the dynamics of the first and second moments are given by

$$\begin{aligned} \frac{d}{dt} \langle S_i \rangle(t) &= \alpha_1 - (\alpha_1 + \alpha_2)\langle S_i \rangle + \beta_1 \langle S_{i\pm1} \rangle, \\ \frac{d}{dt} \langle S_i S_j \rangle(t) &= \alpha_1(\langle S_i \rangle + \langle S_j \rangle) - 2(\alpha_1 + \alpha_2)\langle S_i S_j \rangle + \beta_1(\langle S_{i\pm1} S_j \rangle + \langle S_{j\pm1} S_i \rangle). \end{aligned} \tag{12}$$

We studied the dynamics of the binary-unit network analytically and in simulations. In simulations, the states of all units were updated based on their transition rates in 10 ms time bins.

**Theoretical prediction of noise correlations.** Assuming the network evolved to the equilibrium state, we derived in the continuum limit the steady-state solution for the averaged first moment $S(\infty)$ and quadratic moments $G(d; \infty)$:

$$S(\infty) = \frac{\alpha_1}{\alpha_1 + \alpha_2}, \tag{13}$$

$$G(d; \infty) = [S(\infty)]^2 + S(\infty)(1 - S(\infty)) \exp\left(-\frac{d}{L}\right). \tag{14}$$

Here the dimensionless correlation length $L$ is given by

$$L = \sqrt{\frac{\beta_1}{\alpha_1 + \alpha_2}}, \tag{15}$$

and $d$ is the dimensionless lateral distance measured in units of the lattice constant $\Delta d$.

Using these expressions for the first moment $S(\infty)$ and quadratic moments $G(d; \infty)$, we derived an analytical formula for the noise correlations. Consider a pair of neurons $(\mathbf{x}, i)$ and $(\mathbf{y}, j)$ that are indexed by the lateral positions $\mathbf{x}, \mathbf{y}$ of units to which they belong, and by their indices $i, j$ within these units. Spike-counts $N(\mathbf{x}, i)$ and $N(\mathbf{y}, j)$ of these two neurons are measured in a time-window of duration $T$. The theoretical prediction of noise correlation $r_{\text{sc}}[N(\mathbf{x}, i), N(\mathbf{y}, j)]$ is given by

$$r_{\text{sc}}[N(\mathbf{x}, i), N(\mathbf{y}, j)] = \mathcal{A}(\alpha_1, \alpha_2) \exp\left(-\frac{|\mathbf{x} - \mathbf{y}|}{L}\right). \tag{16}$$

This equation shows that noise correlations decay exponentially with the lateral distance $d = |\mathbf{x} - \mathbf{y}|$, with the decay-rate characterized by the correlation length $L$. The amplitude $\mathcal{A}(\alpha_1, \alpha_2)$ depends on the On-Off transition rates $\alpha_1, \alpha_2$, and on the On/Off firing rates $r_{\text{off}}(\mathbf{x}, i), r_{\text{off}}(\mathbf{y}, j), \Delta r(\mathbf{x}, i), \Delta r(\mathbf{y}, j)$. Specifically,

$$\mathcal{A}(\alpha_1, \alpha_2) = \frac{V(\alpha_1, \alpha_2)\Delta r(\mathbf{x}, i)\Delta r(\mathbf{y}, j)}{\sqrt{\text{Var}[N(\mathbf{x}, i)] \text{Var}[N(\mathbf{y}, j)]}}, \tag{17}$$

$$\begin{aligned} \text{Var}[N(\mathbf{x}, i)] &= (\Delta r(\mathbf{x}, i))^2 V(\alpha_1, \alpha_2) + r_{\text{off}}(\mathbf{x}, i)T + \frac{\alpha_1}{\alpha_1 + \alpha_2} T\Delta r(\mathbf{x}, i), \\ \text{Var}[N(\mathbf{y}, j)] &= (\Delta r(\mathbf{y}, j))^2 V(\alpha_1, \alpha_2) + r_{\text{off}}(\mathbf{y}, j)T + \frac{\alpha_1}{\alpha_1 + \alpha_2} T\Delta r(\mathbf{y}, j), \end{aligned} \tag{18}$$

where

$$V(\alpha_1, \alpha_2) = \frac{2(\alpha_1\alpha_2)}{(\alpha_1 + \alpha_2)^3}\left[T - \frac{1}{\alpha_1 + \alpha_2}\left(1 - \exp(-(\alpha_1 + \alpha_2)T)\right)\right]. \tag{19}$$

The amplitude $\mathcal{A}(\alpha_1, \alpha_2)$ is the theoretical prediction for noise correlations within single columns (in the limit where $d = |\mathbf{x} - \mathbf{y}| \to 0$) used in Fig. 3. In Figs. 4d and 5c, we compute the noise correlation at $d = 0$ (i.e. $\mathcal{A}(\alpha_1, \alpha_2)$) with realistic On and Off firing rates. We sampled 1,000,000 pairs of On and Off firing rates from the distributions estimated by HMM in the V4 data and averaged the noise correlation over all sampled pairs. At distance $d = 0$, the difference between simulations and analytical prediction due to sampling was less than 1%. At distances $d > 0$, we calculate the analytical prediction of noise correlations as the product between the analytical noise correlation at $d = 0$ and the exponential spatial decay factor.

**Reporting summary.** Further information on research design is available in the Nature Research Reporting Summary linked to this article.

## Data availability

All behavioral and electrophysiological data used in this study are available as downloadable files at https://doi.org/10.6084/m9.figshare.16934326.v3. Source data are provided with this paper.

## Code availability

The source code written in Matlab to reproduce results of this study is freely available on GitHub (https://github.com/engellab/Network-models-of-spatiotemporal-On-Off-dynamics).

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

## Acknowledgements

This work was supported by the Swartz Foundation (Y.S.), the NIH grant R01 EB026949 (T.A.E.), the Pershing Square Foundation (T.A.E.), Alfred P. Sloan Foundation Research Fellowship (T.A.E.), and the NIH grant EY014924 (T.M.).

## Author contributions

Y.S., N.A.S., K.B., T.M., and T.A.E. designed the study. N.A.S. and T.M. designed the experiments. N.A.S. performed experiments, spike sorting, and RF measurements. Y.S. and T.A.E. developed data analysis methods and mathematical models. Y.S. analyzed the data, performed analytical calculations and model simulations. Y.S., N.A.S., K.B., T.M., and T.A.E. discussed the findings and wrote the paper.

## Competing interests

The authors declare no competing interests.
