## [Peer Review File · Nature Communications]

Cortical state dynamics and selective attention define the spatial pattern of correlated variability in neocortexREVIEWER COMMENTS

Reviewer #1 (Remarks to the Author):

Shi et al provide a theoretical framework that accounts for both the spatial and attentional modulation of noise correlations. The theoretical idea that on-off dynamics is at the core of the phenomenon is novel, and follows some previous work on the authors by extending it in new ways. The result that noise correlations decrease with distance is well known, but its explanation based on the coupling between nearby columns following somehow independent on-off dynamics is interesting. The authors make strong claims regarding that changes in correlations due to attention are stronger at intermediate distances, but in my opinion this is not clearly supported by the data presented (see below). Overall, I think that the paper provides relevant results and can be suitable for publication provided that the authors address the main comments below:

Major:

-“In the deep layers, the attentional effects on noise correlations were reversed from that in superficial layers (linear regression, one-sided t-test, slope 0.006 ± 0.004 , $p = 0.06$; t-test for slopes superficial versus deep: $p = 0.6 \cdot 10^{-3}$).” The first test seems to be a critical one, but the outcome of the test is that the slope is not significantly positive, if I understand well. Thus, there is no strong evidence in favor that there the changes in correlations with attention increase with distance. However, the authors state in the abstract and several other places that there is an effect. I think that the authors should tone down this result and maybe clearly state that there is a borderline statistical effect, but that there is no strong evidence in favor of an increase of changes of correlations with distance.

-A further discussion between the connection with information-limiting correlations can be useful based on the observation that correlations depend on $\Delta r_i \cdot \Delta r_j$, which pretty much resembles differential correlations in Moreno-Bote et al 2014 and their coarse version in Nogueira et al, J of Neurosc., 2020. In particular, noise correlations in the direction of the tuning harm information, but not if they are not aligned with tuning. Thus, a discussion about whether the changes in correlations due to attention can change or modify their alignment with tuning will be relevant to understand the possible effects of attention on coding.

-How is the model comparison between a single and two-phases HHM performed? Are the explained variances in Fig. 1c cross-validated? It would be interesting to compare the single and two-phases HHMs by showing which of the two models account best for cross-validated explained variance –that is, employing data not used in the fit of the model.

-“Whereas all neurons represented by a single unit in the network follow the same shared sequence of On-Off phases, neurons represented by different units follow their respective On-Off sequences.” Can this be illustrated with some spike rasters? It seems like a central prediction, but beyond the decrease of correlations, not further data is presented

Minor:

-The model assumes that attention introduces an external excitatory input to the column, but there is not discussion about the validity or realism of this assumption.

-It would be relevant to show distributions of on and off durations as supplementary material

-The increase of on-durations with attention could be presented as a supplementary figure

-Fig. 3b would be better represented as correlation_{ij} vs $\Delta r_i \cdot \Delta r_j$ in the form of a scatter plot

-“Despite substantial changes in many pairs, the average change in noise correlations within columns is near zero (Fig. 1b)...” Fig. 1b does not seem to show the distribution of correlations, which could be shown as a SI figure.

Reviewer #2 (Remarks to the Author):

Review of NCOMMS-20-47313-T : Cortical state dynamics and selective attention define the spatial pattern of correlated variability in neocortex by Shi et al.

The goal of this paper is to understand how spatial attention shapes spatial patterns of noise correlation in this sensory area. To this end, it combines analysis of extracellular recordings in area V4 in monkey performing a spatial attention task with modeling and theory.

The recordings show that the cortical network undergoes random transitions between states of low activity (off-state) and states of high-activity (on-state) with transition interval durations exponentially distributed. The key conclusion of the paper is that the On-Off dynamics are a dominant source of noise correlation in the visual cortex. These dynamics can predict the magnitude of these correlations, their lateral spatial dependence as well as their modulation by attention. This is explained theoretically in a minimal network model of V4 comprising bistable units.

The model makes a testable prediction that the largest changes of noise correlations driven by attention occur at intermediate lateral distances. The authors report experimental data which suggest that this is the case in upper layer in V4.

In conclusion, the authors argue that the reported attentional driven reduction of noise correlations may be a major contributor to the improved psychophysical performance during attention.

General:

Overall, this paper is very interesting. The theoretical model is elegant by its simplicity and by the possibility to reduce it to a network of interacting binary elements amenable to analytical treatment. The mathematical analysis of the reduced model is nice and the comparison of the results with the data very instructive.

Specific comments:

I suggest to the authors to take into account the following issues and comments.

1) The description of the experiments:

The experimental protocol is briefly described in the first section of the Results and with more detail in the section "Behavior and electrophysiology" in the Methods. The protocol is identical to the one in Ref. 22. Adding in the text or in the supplementary information a figure similar to Fig. 1 B & D of the latter paper will be useful to the reader.

2) On-Off dynamics (spikes) & Up-Down states (voltage):

I think that the "general reader" will be interested in learning a bit about the connection between the on/off and up/down transitions as well as on the experimental conditions (e.g. anesthetized or awake) under which they are observed experimentally.

3) Mechanism of on/off dynamics:

In the theoretical model developed here the on/off dynamics stem from a bi-stability of the local cortical network dynamics. Theoretical neuroscientist, especially those with a physics background, will be interested by a short discussion about the mechanisms of on/off dynamics proposed in previous works.

4) Noise in the model:

The noise is introduced in the model via the dynamical equation of the adaptation variable and not in the equation satisfied by the activity variable. As a result, the noise can drive the on/off transitions in the limit $\epsilon \rightarrow 0$.

What will happen if the dynamics of the activity and of the adaptation have comparable time constant? Can on/off transitions be driven by the noise if it is introduced only in the dynamics of

the activity?

5)Page 2: The author writes "We hypothesized that heterogeneous changes in noise correlations arise from the modulation On-Off dynamics propagating through spatially structured cortical connectivity" and in Page 3 they write "The key mechanism in our model is On-Off dynamics that propagate across columns to form spatiotemporal population activity which shapes the structure of noise correlations. Cortical activity propagates laterally as traveling waves across different brain states and behavioral conditions and wave-like propagation of spontaneous activity fluctuations is observed in the visual cortex of awake, behaving primates."

In the full-model studied here the noise correlations decrease exponentially with distance. Why do the authors think that their experimental and modeling results stem from the propagation of waves of neuronal activity? I think that this is in fact not correct (think for instance about the Ising model).

6)Page 4-5, section "Attentional modulation of noise correlations within columns"

The last paragraph of this section deals with noise correlation between columns. It seems to be misplaced.

7)The model:

-The network which is simulated and analytically studied has only one layer. Why including a second layer in its description (text and Fig. 2a)?

-The neurons within one cortical column are not explicitly modeled. Therefore, an implicit assumption of the model is that the noise which induces the transitions between on/off states is correlated on a scale of the order of the size of a column. If I am correct, this should be mentioned and justified. If I am wrong, this should be clarified.

-The reduced model has some resemblance with a 2-D Ising model. I suggest to include a brief discussion about that as well as about the differences/similarities in their "physics". This will be interesting to the general physicist reader.

-The mathematical analysis of the reduced model is performed in the continuum limit. In statistical mechanics, taking this limit is valid only near criticality where the correlation length diverges. To what extent the continuum limit can be justified in the present model?

8)Page 9: "This analytical result agrees well with simulations of the full dynamical-system network model (Fig. 4d)"

The analytics and the simulations give very different results for $d=0$. What is the origin of this discrepancy?

9)Page 10: "In simulations, excitatory attentional inputs produce a faster decay of noise correlations with lateral distance, which corresponds to a shorter correlation length ($L_{att} < L_{ctl}$, Fig. 5c). Due to this faster spatial decay, noise correlations at intermediate lateral distances (finite $d > 0$) are considerably lower in attention relative to control condition, even though changes of noise-correlations within columns ($d = 0$) are small."

In Fig. 5c, the red and black data points and curves are very close to each other. I do not understand why the author says this figure shows that there is a "faster decay of noise correlations with distance and "considerably lower" noise correlations at intermediate distances, unless d is normalized to L . If this is the case this should be mentioned.

It would be nice to add a panel which compares the dependence of L with L_{att} in the simulation with the analytical prediction.

10)Page 11, bottom and Fig. 6d,e:

The authors say: "Changes of noise correlations were smallest at very short lateral distances and became progressively larger at longer distances, with opposite trends in the superficial and deep layers (Fig. 6d,e)." I do not see such an effect in Fig. 6d,e. Please clarify.

11)Page 13: "The model accurately captures attention-related changes of noise correlations in our data." I think that saying "accurately" is too optimistic. I suggest "qualitatively".

12)Page 13: "Moreover, it makes a testable prediction that the largest changes of noise correlations occur at intermediate lateral distances. Consistent with this prediction, we found that the magnitude of noise-correlation changes, gradually increased with lateral distance in our laminar recordings." Here again, the authors seem to me over-optimistic. It is difficult to appreciate the non-monotonicity in Fig. 6d in the absence of error bars for the data point representing DeltaNC across columns. The data in Fig. 6e are too noisy to really draw the conclusion that DeltaNC increases with distance.

Minor:

-In several places: It should be "adaptation" instead of "adaption"
-Ref. 29 is incomplete

Reviewer #3 (Remarks to the Author):

This manuscript explores the mechanisms by which attention affects single neuron variability and pairwise correlations in the visual cortex. This topic has been of substantial interest in the past ~10 years, with numerous reports showing how attention alters correlations and with some network models proposed to explain the observed effects. Here the authors report small effects of attention on correlations with a cortical column and larger effects when the electrode penetration samples neurons in different columns. These effects—and other previous studies—can be reconciled in an 'on/off' model in which correlations arise from shared state transitions (within a column), whose spread across columns depends on the attentional signal. The proposed explanation is quite different from other recently published models (mostly the work of Brent Doiron and colleagues, cited).

This is a high quality study on a topic of wide interest. The presentation is clear. Because in the data the sampling across space is limited and because the effects in superficial and deep layers are different, I am not entirely convinced of the claims. But the study, nevertheless, contains a number of interesting points and provides an alternative explanation for some widely studied effects (attention-induced decorrelations). This is scholarly work. It is just that more data will be needed in future work to adjudicate between the competing models.

Although I am supportive of publication, I do have a few questions, comments, and suggestions.

Comments:

1. The Fano factors shown in Fig 3a look unusually high (up to 9 for a counting window of a few hundred milliseconds). Please explain or justify this apparent discrepancy with prior work.
2. I did not understand why the fits for the 1 phase model was much worse than for the 2 phase model (Fig 1b), when the 1 phase model was the better of the two. Please explain/justify.
3. I was also puzzled by the statement that attention reduced Fanos for 1 phase recordings similarly as for 2 phase (Supp Table, 1,2; main text page 5). If the attention works by biasing to the On state (for $I_{att} > 0$), why is there an equally large change in Fanos in sessions in which those fluctuations are not evident. Put another way, why do Fanos change at all in 1 phase sessions?
4. Similarly, if correlation are due to shared On/Off transitions (and depend on $r_{on} - r_{off}$), why

are correlations for 1 phase recordings not equal to 0? This suggests there is some other mechanism (important, in that the correlations for 1 phase recordings are an appreciable fraction of those measured for 2 phase recordings) that generates correlations, in addition to the mechanism suggested by the authors. This issue should be addressed in the Discussion.

5. On page 12, the authors discuss the effect of reducing correlations on population information. But it is now quite well accepted that changes in mean correlations do not indicate much about information (see Moreno-Bote et al., 2015, NN and others) so these statements should be revised.

6. I was a bit thrown by the use of 'On/Off' as a term for the two states. In fact, in the 'Off' state, the measured neuronal responses are still appreciable. This is in contrast to a rich literature on On/Off states (work of McCormick and others) that has shown almost total quiescence in the Off state. It might be less confusing (with respect to this prior literature) to refer to the effects here as 'Low' and 'High rate' states. At the very least, if the authors stick to the current terminology, they should explain that this same term means something quite different in other systems/network neuroscience literature.

Point-by-point responses.

Reviewer #1:

Shi et al provide a theoretical framework that accounts for both the spatial and attentional modulation of noise correlations. The theoretical idea that on-off dynamics is at the core of the phenomenon is novel, and follows some previous work on the authors by extending it in new ways. The result that noise correlations decrease with distance is well known, but its explanation based on the coupling between nearby columns following somehow independent on-off dynamics is interesting. The authors make strong claims regarding that changes in correlations due to attention are stronger at intermediate distances, but in my opinion this is not clearly supported by the data presented (see below). Overall, I think that the paper provides relevant results and can be suitable for publication provided that the authors address the main comments below:

Reply:

We thank the reviewer for the positive assessment and suggestions that helped us improve the paper.

1) *"In the deep layers, the attentional effects on noise correlations were reversed from that in superficial layers (linear regression, one-sided t-test, slope 0.006 ± 0.004 , $p = 0.06$; t-test for slopes superficial versus deep: $p = 0.6 \cdot 10^{-3}$)." The first test seems to be a critical one, but the outcome of the test is that the slope is not significantly positive, if I understand well. Thus, there is no strong evidence in favor that there the changes in correlations with attention increase with distance. However, the authors state in the abstract and several other places that there is an effect. I think that the authors should tone down this result and maybe clearly state that there is a borderline statistical effect, but that there is no strong evidence in favor of an increase of changes of correlations with distance.*

Reply:

In our data, noise correlations decreased in superficial layers and increased in deep layers during attention. In superficial layers, the decrease in noise correlations was significantly greater at longer distances (linear regression, one-sided t-test, slope -0.017 ± 0.004 , $p < 10^{-3}$, Fig. 6d). In deep layers, the increase in noise correlations was also greater at longer distances, but as the reviewer points out, this effect did not reach statistical significance. We revised the main text and the abstract to clarify that we confirm model prediction by showing that the largest changes in correlations occur at intermediate lateral distances in superficial layers. Deep layers, which differ significantly in sign from superficial layers, show a moderate trend with a borderline statistical effect.

2) *A further discussion between the connection with information-limiting correlations can be useful based on the observation that correlations depend on $\Delta r_i \cdot \Delta r_j$, which pretty much resembles differential correlations in Moreno-Bote et al 2014 and their coarse version in Nogueira et al, J of Neurosc., 2020. In particular, noise correlations in the direction of the tuning harm information, but not if they are not aligned with tuning. Thus, a discussion about whether the changes in correlations due to attention can change or modify their alignment with tuning will be relevant to understand the possible effects of attention on coding.*

Reply:

Following the reviewer's suggestion, we analyzed the relationship between On-Off dynamics and differential correlations. In our model, assuming that stimulus does not change the statistics of On-Off dynamics and that changes in the On-Off dynamics do not affect stimulus tuning, we found that the On-Off dynamics influence the strength of differential correlations and thus affect the saturation level of information. Specifically, the linear Fisher information is monotonically decreasing with the correlation length. Hence, a reduction of the correlation length leads to an increase in stimulus information, as we observed in superficial cortical layers during attention. However, the On-Off dynamics and stimulus tuning are likely interdependent in the cortical circuitry, where both arise from the same structured connectivity. In future work, deeper understanding of how On-Off fluctuations impact sensory coding will be possible using models with connectivity that supports stimulus tuning in addition to spatial receptive fields. We describe these results in new Supplementary Note 4 and briefly summarize them in the Discussion section.

We would also like to clarify that although the term $\Delta r_i \Delta r_j$ resembles differential correlations in Nogueira et al., its meaning is different. In our case, Δr_i represents the difference between the On and Off firing rates for a fixed stimulus, whereas differential correlations are proportional to the difference in mean firing rates elicited by two different stimuli.

3) *How is the model comparison between a single and two-phases HHM performed? Are the explained variances in Fig. 1c cross-validated? It would be interesting to compare the single and two-phases HHMs by showing which of the two models account best for cross-validated explained variance –that is, employing data not used in the fit of the model.*

Reply:

The cross-validation procedure was the same as in our previous work (see Supplementary Fig. 5 in Engel et al., Science 2016). We computed average cross-validation error for HHMs with n phases ($n = 1, \dots, 8$), normalized by the average cross-validation error of the HHM with 1 phase. For each recording session, the 4-fold cross-validation error was computed in 200 ms windows for each condition (4 attention conditions \times 8 stimulus orientations), and then averaged across all channels, conditions and cross-validation folds. For most recordings, addition of the second phase greatly reduced cross-validation error compared to the single-phase HHM, whereas adding more phases resulted in only marginal improvements: the error curves display an elbow at $n = 2$, suggesting that 2-phase HHM is the most parsimonious model for our data. For some recording sessions, HHMs with $n > 1$ phases did not perform better than the 1-phase HHM and the error curves did not exhibit a kink for these recordings. We defined these to be one-phase recordings. We now included the description of model comparison in the Methods section of the main text.

4) *“Whereas all neurons represented by a single unit in the network follow the same shared sequence of On-Off phases, neurons represented by different units follow their respective On-Off sequences.” Can this be illustrated with some spike rasters? It seems like a central prediction, but beyond the decrease of correlations, not further data is presented*

Reply:

In our laminar recordings, most neurons have very small lateral separation (i.e. are located within a single column). These neurons follow approximately the same sequence of On-Off phases (Fig. 1b), similar to a single unit in the model. We also obtained five recording sessions in one monkey during a fixation task, where we inserted the linear probe at a slight angle, so that neurons show larger lateral shifts between their receptive fields (larger lateral separation). In these sessions, we indeed can observe On-Off phases that occur synchronously only on a subset of adjacent channels and propagate across channels over time. We show such example recording in new Supplementary Fig. 10. In the example raster plots (Supplementary Fig. 10b), we can see that On-Off phases are less synchronized across distant channels, suggesting that neurons in different columns follow their respective On-Off sequences. We provide Supplementary Fig. 10 as an illustration supporting our model. However, the current dataset is not sufficient for a more systematic analysis of the coordination of On-Off dynamics across columns, which we defer to future work.

5) *The model assumes that attention introduces an external excitatory input to the column, but there is not discussion about the validity or realism of this assumption.*

Reply:

We added a discussion of validity of this assumption in Discussion. Top-down projections from frontal cortical areas, especially Frontal Eye Fields (FEF) can provide temporally and spatially precise inputs to drive fast and local modulation of On-Off dynamics in the visual cortex. Most FEF projections to V4 target pyramidal neurons [Anderson et al, J. Neurosci. 31, 10872 (2011)], in agreement with our model where reduction of noise correlations in superficial layers is driven by external excitatory inputs.

6) *It would be relevant to show distributions of on and off durations as supplementary material.*

Reply:

We included a new Supplementary Fig. 4 showing distributions of On and Off durations for an example recording.

7) *The increase of on-durations with attention could be presented as a supplementary figure.*

Reply:

The increase of On-durations with attention has been shown in our previous work: Fig. 3 in Engel, et al. Science 354, 1140 (2016), see also van Kempen et al., Neuron 109, 894 (2021). We also included a new Supplementary Fig. 5 showing the increase of On-durations with attention.

8) *Fig. 3b would be better represented as correlation_{ij} vs Delta_{r_i}*Delta_{r_j} in the form of a scatter plot.*

Reply:

In Fig. 3b, we included a scatter plot of noise correlations versus $\sqrt{\Delta r_i \Delta r_j}$, which has the units of firing rate (Hz) and thus is more interpretable.

9) *“Despite substantial changes in many pairs, the average change in noise correlations within columns is near zero (Fig. 1b).” Fig. 1b does not seem to show the distribution of correlations, which could be shown as a SI figure.*

Reply:

We added a new Supplementary Fig. 8 showing the distribution of noise correlations and their changes during attention and revised the main text accordingly.

Reviewer #2:

The goal is of this paper is to understand how spatial attention shapes spatial patterns of noise correlation in this sensory area. To this end, it combines analysis of extracellular recordings in area V4 in monkey performing a spatial attention task with modeling and theory. The recordings show that the cortical network undergoes random transitions between states of low activity (off-state) and states of high-activity (on-state) with transition interval durations exponentially distributed. The key conclusion of the paper is that the On-Off dynamics are a dominant source of noise correlation in the visual cortex. These dynamics can predict the magnitude of these correlations, their lateral spatial dependence as well as their modulation by attention. This is explained theoretically in a minimal network model of V4 comprising bistable units. The model makes a testable prediction that the largest changes of noise correlations driven by attention occur at intermediate lateral distances. The authors report experimental data which suggest that this is the case in upper layer in V4. In conclusion, the authors argue that the reported attentional driven reduction of noise correlations may be a major contributor to the improved psychophysical performance during attention.

General:

Overall, this paper is very interesting. The theoretical model is elegant by its simplicity and by the possibility to reduce it to a network of interacting binary elements amenable to analytical treatment. The mathematical analysis of the reduced model is nice and the comparison of the results with the data very instructive.

Reply:

We thank the reviewer for this generous assessment and for many insightful comments that helped us improve the paper.

1) The description of the experiments: The experimental protocol is briefly described in the first section of the Results and with more detail in the section “Behavior and electrophysiology” in the Methods. The protocol is identical to the one in Ref. 22. Adding in the text or in the supplementary information a figure similar to Fig. 1 B & D of the latter paper will be useful to the reader.

Reply:

We added new Supplementary Fig. 1 describing the behavioral task.

2) On-Off dynamics (spikes) & Up-Down states (voltage): I think that the “general reader” will be interested in learning a bit about the connection between the on/off and up/down transitions as well as on the experimental conditions (e.g. anesthetized or awake) under which they are observed experimentally.

Reply:

We added a brief discussion of the On-Off dynamics versus Up-Down states in Discussion. On-Off dynamics refer to spontaneous transitions between episodes of vigorous (On) and faint (Off) spiking that occur synchronously across layers in the cortex of behaving monkeys. Up-Down states were originally used to refer to the two modes of the bimodal distribution of membrane potentials, and now also used to refer to spiking (Up) and silent (Down) phases of population activity during slow-wave sleep and anesthesia.

3) Mechanism of on/off dynamics: In the theoretical model developed here the on/off dynamics stem from a bi-stability of the local cortical network dynamics. Theoretical neuroscientist, especially those with a physics background, will be interested by a short discussion about the mechanisms of on/off dynamics proposed in previous works.

Reply:

We added a description of mechanisms proposed previously for Up-Down dynamics in Discussion. Our model of On-Off dynamics is based on bistability, which is consistent with exponential distributions of On and Off-episode durations in behaving monkeys. Similar models with slow negative feedback were used to reproduce Up-Down dynamics. This mechanism can generate slow alternations between high and low firing rate via several dynamical regimes. In particular, Up-Down dynamics were found to be bistable during anesthesia [D. Jercog et al. ELife 6, e22425 (2017)] and excitable during slow wave sleep (a single stable fixed point from which suprathreshold fluctuations induce large transient events) [Levenstein et al, Nat. Commun. 10, 2478 (2019)].

4) *Noise in the model: The noise is introduced in the model via the dynamical equation of the adaptation variable and not in the equation satisfied by the activity variable. As a result, the noise can drive the on/off transitions in the limit $\epsilon \rightarrow 0$. What will happen if the dynamics of the activity and of the adaptation have comparable time constant? Can on/off transitions be driven by the noise if it is introduced only in the dynamics of the activity?*

Reply:

In our model, the parameter $\epsilon \ll 1$ ensures that the time constant is slower for the adaptation variable than for the firing-rate variable. The separation of timescales restricts the system's dynamics in the phase space to a narrow region around the left and right branches of the firing-rate nullcline and the two lines connecting them [Lindner & Schimansky-Geier, Phys. Rev. E 60, 7270 (1999)]. This behavior accounts for the timescale separation in the On-Off dynamics in the monkey cortex: in the data, transitions between the On and Off phases occur much faster than the average dwelling times in these On and Off phases [Engel et al., Science 354, 1140 (2016)]. In the model, increasing ϵ with fixed noise strength Q reduces the transition rate between fixed points, because the variance of firing-rate fluctuations around the fixed point is proportional to Q/ϵ^2 (new Supplementary Note 2.2.3 and Supplementary Fig. 12). On-Off transitions in the model can also be driven by noise if it is introduced only in the dynamics of the firing-rate variable (new Supplementary Note 2.2.4 and Supplementary Fig. 13).

5) *Page 2: The author writes “We hypothesized that heterogeneous changes in noise correlations arise from the modulation On-Off dynamics propagating through spatially structured cortical connectivity” and in Page 3 they write “The key mechanism in our model is On-Off dynamics that propagate across columns to form spatiotemporal population activity which shapes the structure of noise correlations. Cortical activity propagates laterally as traveling waves across different brain states and behavioral conditions and wave-like propagation of spontaneous activity fluctuations is observed in the visual cortex of awake, behaving primates.” In the full-model studied here the noise correlations decrease exponentially with distance. Why do the authors think that their experimental and modeling results stem from the propagation of waves of neuronal activity? I think that this in fact not correct (think for instance about the Ising model).*

Reply:

Exponential decay of noise correlations with distance is not in conflict with propagation of neural activity across columns. In our model, spontaneous activity fluctuations form spatial clusters which propagate laterally in a pattern we call “local irregular waves”. Due to stochasticity of dynamics, the activity clusters do not propagate coherently across the entire network, but travel only locally until they fade or merge with other clusters. Local irregular waves differ from global traveling waves, in which a wave can propagate coherently across the entire network and most neurons equally participate in each wave [Engel & Steinmetz, Current Opinion in Neurobiology 58, 181 (2019)]. In our model, the scale of spatial clusters is restricted to a few columns and defines the spatial decay constant of noise correlations (correlation length). In simulations, the activity exhibits local irregular waves with the size defined by correlation length, similar to the disordered phase of Ising model above the critical temperature. We clarified this issue in the revised Discussion and included a new Supplementary Movie 1 to show an example of the “local irregular waves” pattern in our model.

6) *Page 4-5, section “Attentional modulation of noise correlations within columns” The last paragraph of this section deals with noise correlation between columns. It seems to be misplaced.*

Reply:

We thank the reviewer for this comment. This paragraph compares noise correlations in our columnar recordings to results of previous studies that recorded from different columns using Utah arrays. We also report the modulation of Fano factor in our columnar recordings and compare it to previous work. In this sense, this paragraph still deals with noise correlations within columns. We therefore decided to leave it in this sub-section.

7) *The model:*

-The network which is simulated and analytically studied has only one layer. Why including a second layer in its description (text and Fig. 2a)?

Reply:

We revised the model description on page 5 and the caption of Fig. 2 to clarify why we included two layers in our network model. Since attentional modulation differs between superficial and deeper layers, we modeled each layer as a separate network. Each network receives different attentional inputs to account for differential changes in noise correlations in superficial versus deep layers.

-The neurons within one cortical column are not explicitly modeled. Therefore, an implicit assumption of the model is that the noise which induces the transitions between on/off states is correlated on a scale of the order of the size of a column. If I am correct, this should be mentioned and justified. If I am wrong, this should be clarified.

Reply:

We clarified this issue in Methods. The noise term in our model drives stochastic transitions between the On and Off phases in single columns. This phenomenological noise term models fluctuations in population activity due to internal spiking noise and/or external stochastic inputs. Spiking noise can arise from finite-size fluctuations [Schwalger et al., PLoS Comp Biol 13, e1005507 (2017)] as well as from biophysical sources, such as stochasticity of ion channels or synaptic release. External inputs correlated on the spatial scale of a column could also probabilistically drive On-Off transitions, for example, inputs related to microsaccades [Engel et al., Science 354, 1140 (2016)]. The term $\xi(t)$ models the net effect of various biological noise sources.

-The reduced model has some resemblance with a 2-D Ising model. I suggest to include a brief discussion about that as well as about the differences/similarities in their “physics”. This will be interesting to the general physicist reader.

Reply:

We added new Supplementary Note 2.4 and a brief discussion in Methods, where we compare the dynamics of our binary-unit model and the 2-D Ising model. In the binary-unit model, we describe transitions of binary units using the Glauber dynamics formalism [Glauber, J. Math. Phys. 4, 294 (1963)]. In statistical physics, Glauber dynamics are used to simulate the dynamics of the Ising model, which describes the spin system not just at equilibrium but also during transitional stages. We find that for the 2-D Ising model, the transition rate can be written in the same form as in our binary-unit model. However, the detailed balance condition of thermodynamics constrains the ratio of transition rate parameters of the Ising model, which are related to interaction strength in the Hamiltonian and temperature. In general, the detailed balance condition is not satisfied in the binary-unit model, so its dynamics are different from the 2-D Ising model.

-The mathematical analysis of the reduced model is performed in the continuum limit. In statistical mechanics, taking this limit is valid only near criticality where the correlation length diverges. To what extent the continuum limit can be justified in the present model?

Reply:

We use the continuum limit to derive the analytical form of correlation function and correlation length. To confirm that the continuum limit is valid, we compared the analytical prediction of the correlation length to simulation result (new Supplementary Fig. 11a). We found overall the simulation results were consistent with the analytical prediction, which validates the approximation of continuum limit.

8) Page 9: “This analytical result agrees well with simulations of the full dynamical-system network model (Fig. 4d)” The analytics and the simulations give very different results for $d=0$. What is the origin of this discrepancy?

Reply:

We thank the reviewer for pointing out this issue. We reexamined the results and found that the discrepancy was due to insufficient sampling of firing rates. The magnitude of noise correlations depends on the firing rates during On and Off phases, even when the On-Off sequence is fixed (Eq. 96 in Supplementary Information). In both simulations and analytical computations, we used On and Off firing rates of neurons randomly sampled from the distributions estimated by HMM in the V4 data. Previously, we used different random samples of firing rates in simulations and analytical calculations with a relatively small sample size. The discrepancy between simulations and analytical predictions in Fig. 4d was due to insufficient sample size of firing rates, which resulted in high variability between two samples. We corrected this issue by increasing the sample size of On and Off firing rates. In simulations, we repeated sampling 10 times (Supplementary Note 2.2.2). To confirm that this

sample size was sufficient, we verified that the results did not change when we further increased the sample size. For analytical predictions of noise correlation at distance $d=0$, we now sample 1,000,000 pairs of On and Off firing rates and average noise correlations over all sampled pairs. With this large sample size, the difference between simulations and analytical prediction was less than 1%. At distances $d>0$, we calculate the analytical prediction of noise correlations as the product between the analytical noise correlation at $d=0$ and the spatial decay factor. We revised the description of the sampling procedure in Methods and Supplementary Note 2.2.2. We updated Fig. 4d and Fig. 5c using the larger sample size. After increasing the sample size, the simulation results tightly overlap with the analytical predictions (deviation is less than 10%).

9) Page 10: *“In simulations, excitatory attentional inputs produce a faster decay of noise correlations with lateral distance, which corresponds to a shorter correlation length ($L_{att} < L_{ctl}$, Fig. 5c). Due to this faster spatial decay, noise correlations at intermediate lateral distances (finite $d > 0$) are considerably lower in attention relative to control condition, even though changes of noise-correlations within columns ($d = 0$) are small.” In Fig. 5c, the red and black data points and curves are very close to each other. I do not understand why the author says this figure shows that there is a “faster decay of noise correlations with distance and “considerably lower” noise correlations at intermediate distances, unless d is normalized to L . If this is the case this should be mentioned. It would be nice to add a panel which compares the dependence of L with L_{att} in the simulation with the analytical prediction.*

Reply:

The difference of noise correlations between attention and control conditions is indeed small (both in model and experiment), which makes it hard to see the change in the correlation length in Fig. 5c. When we fit simulation data with an exponential function $a \cdot \exp(-b \cdot x)$ (x is the distance between a pair of units), we find a significant decrease of the correlation length with attentional input (new Supplementary Fig. 11b). Therefore, we state that there is a “faster decay of noise correlations with distance and “considerably lower” noise correlations at intermediate distances. To better visualize the faster decay of noise correlations, we included an inset in Fig. 5c showing the noise correlations in attention and control conditions on a linear-logarithmic scale. In our analysis, distance d and correlation length L are both normalized by the lattice constant. Following the reviewer’s suggestion, we included a new Supplementary Fig. 11b, which compares the dependence of L on the attentional input in the simulations and the analytical prediction.

10) Page 11, bottom and Fig. 6d,e: *The authors say: “Changes of noise correlations were smallest at very short lateral distances and became progressively larger at longer distances, with opposite trends in the superficial and deep layers (Fig. 6d,e).” I do not see such an effect in Fig. 6d,e. Please clarify.*

Reply:

We revised the main text and the caption of Fig. 6d,e to clarify that we observed larger changes in noise correlations with increasing distance in superficial layers, while in deep layers, we observed a moderate increasing trend with a borderline statistical effect. In the superficial layers, noise correlations decreased during attention, with greater reduction at longer distances (linear regression, one-sided t-test, slope -0.017 ± 0.004 , $p < 10^{-3}$). In deep layers, the changes in noise correlations showed a moderate increasing trend with distance with a borderline statistical effect (linear regression, one-sided t-test, slope 0.006 ± 0.004 , $p = 0.06$).

11) Page 13: *“The model accurately captures attention-related changes of noise correlations in our data.” I think that saying “accurately” is too optimistic. I suggest “qualitatively”.*

Reply:

We agree and we modified the text accordingly.

12) Page 13: *“Moreover, it makes a testable prediction that the largest changes of noise correlations occur at intermediate lateral distances. Consistent with this prediction, we found that the magnitude of noise-correlation changes, gradually increased with lateral distance in our laminar recordings.” Here again, the authors seem to me over-optimistic. It is difficult to appreciate the non-monotonicity in Fig. 6d in the absence of error bars for the data point representing ΔNC across columns. The data in Fig. 6e are too noisy to really draw the conclusion that ΔNC increases with distance.*

Reply:

We agree that it is difficult to appreciate non-monotonicity based solely on the data point for ΔNC across columns (Utah array) in the absence of error bars. We would like to clarify that we show this data point to facilitate comparison with previous studies of noise-correlation changes using Utah arrays, which sample

primarily from superficial cortical layers. Similar to Utah-array recordings, we find that in superficial layers (Fig. 6d) noise correlations decrease during attention at intermediate lateral distances (RF-center distance $\sim 2-4$ dva), whereas ΔNC approaches zero at zero distance (RF-center distance = 0 dva). Thus, in superficial layers, changes in noise correlations depend on lateral distance (Fig. 6d, linear regression, one-sided t-test, slope -0.017 ± 0.004 , $p < 10^{-3}$). We clarified this issue in the revised the main text. We also agree that the data in Fig. 6e are noisy and the positive trend does not reach statistical significance. We modified the main text accordingly.

13) *Minor:*

-In several places: It should be “adaptation” instead of “adaption”

-Ref. 29 is incomplete

Reply:

We thank the reviewer for pointing out these typos and incomplete reference. We have made corrections accordingly.

Reviewer #3:

This manuscript explores the mechanisms by which attention affects single neuron variability and pairwise correlations in the visual cortex. This topic has been of substantial interest in the past ~ 10 years, with numerous reports showing how attention alters correlations and with some network models proposed to explain the observed effects. Here the authors report small effects of attention on correlations with a cortical column and larger effects when the electrode penetration samples neurons in different columns. These effects—and other previous studies—can be reconciled in an ‘on/off’ model in which correlations arise from shared state transitions (within a column), whose spread across columns depends on the attentional signal. The proposed explanation is quite different from other recently published models (mostly the work of Brent Doiron and colleagues, cited). This is a high quality study on a topic of wide interest. The presentation is clear. Because in the data the sampling across space is limited and because the effects in superficial and deep layers are different, I am not entirely convinced of the claims. But the study, nevertheless, contains a number of interesting points and provides an alternative explanation for some widely studied effects (attention-induced decorrelations). This is scholarly work. It is just that more data will be needed in future work to adjudicate between the competing models. Although I am supportive of publication, I do have a few questions, comments, and suggestions.

Reply:

We thank the reviewer for the overall positive feedback and for detailed comments that helped us further improve the paper.

1) *The Fano factors shown in Fig 3a look unusually high (up to 9 for a counting window of a few hundred milliseconds). Please explain or justify this apparent discrepancy with prior work.*

Reply:

In our dataset, the median Fano factor of MUA is 1.8 for a 200 ms time-window, which is consistent with prior work. We included the histogram of Fano factor in the new Supplementary Fig. 6a. The histogram shows that Fano factor is less than 2 for the majority of units, while some units show high Fano factor. Based on our analytical prediction of Fano factor for two-phase recordings (Supplementary Information, Eq. 5-8), Fano factor is proportional to the On-Off firing rate difference ($\Delta r = r_{on} - r_{off}$). While the median On-Off firing-rate difference is about 38 Hz, for some multi-units it is higher than 150Hz (Supplementary Fig. 6b), leading to high Fano factor. Thus, although some units in our dataset showed high Fano factor (these are units with the largest On-Off firing rate difference, which are typically MUA as in Fig 3a), the average Fano factor in our data was similar to previous reports. We clarified this issue in the revised main text.

2) *I did not understand why the fits for the 1 phase model was much worse than for the 2 phase model (Fig 1b), when the 1 phase model was the better of the two. Please explain/justify.*

Reply:

We think the reviewer refers to Fig 1c, where we show variance explained by a two-phase HMM across two-phase and one-phase recordings. We classified a recording as a one-phase recording if the two-phase HMM did not provide a substantial improvement over the one-phase HMM (see the revised Methods section). As the reviewer points out, the two-phase HMM accounts for a smaller fraction of total variance in spike counts in one-phase compared to two-phase recordings. The reason for this difference is that the total variance of spike counts consists of two contributions: firing-rate variability and the point-process (spiking) variability [Churchland et al., Neuron 69, 818 (2011)]. HMM models the firing-rate variability as transitions between states, and the one-phase

HMM corresponds to a constant firing rate. HMM models spiking variability as a Poisson process, which is uncorrelated across time bins when conditioned on the firing rate. Poisson variability is irreducible, i.e. it cannot be predicted with an HMM. Therefore, HMM can only account for the firing-rate variability. In two-phase recordings, the firing-rate variability constitutes a substantial fraction of the total spike-count variance, and therefore HMM accounts a large amount of the total variance. In one-phase recordings, the firing-rate variability is very small and the largest fraction of the total spike-count variance is due to Poisson variability. Since Poisson variability is not predictable, the amount of total variance explained by HMM is low. We explained this effect in the revised main text.

3) *I was also puzzled by the statement that attention reduced Fanos for 1 phase recordings similarly as for 2 phase (Supp Table, 1,2; main text page 5). If the attention works by biasing to the On state (for $I_{att} > 0$), why is there an equally large change in Fanos in sessions in which those fluctuations are not evident. Put another way, why do Fanos change at all in 1 phase sessions?*

Reply:

We agree that this is an important question and we addressed it in the revised Discussion. Indeed, the presence of nonzero Fano factor and noise correlation effects in one-phase recordings indicates that other sources contribute to neural variability besides On-Off dynamics. Since in one-phase recordings, noise correlations did not depend on distance (Fig. 4c), these other variability sources may be more global and uniform within a cortical area, such as fluctuations in neural excitability related to arousal. Moreover, Fano factor and noise correlations in one-phase recordings were modulated during attention, suggesting that the additional variability sources also interact with attentional mechanisms, producing spatially uniform changes in correlated variability (Fig. 6f).

4) *Similarly, if correlation are due to shared On/Off transitions (and depend on $r_{on-r_{off}}$), why are correlations for 1 phase recordings not equal to 0? This suggests there is some other mechanism (important, in that the correlations for 1 phase recordings are an appreciable fraction of those measured for 2 phase recordings) that generates correlations, in addition to the mechanism suggested by the authors. This issue should be addressed in the Discussion.*

Reply:

We agree that this is an important point, which we now address in the revised Discussion together with the question 3 about Fano factor (see reply above to the question 3). Non-zero noise correlations in one-phase recordings indicate that besides On-Off dynamics other sources of variability contribute to noise correlations. Since in one-phase recordings noise correlations and their attentional modulation did not depend on lateral distance, these other variability sources are likely more global.

5) *On page 12, the authors discuss the effect of reducing correlations on population information. But it is now quite well accepted that changes in mean correlations do not indicate much about information (see Moreno-Bote et al., 2015, NN and others) so these statements should be revised.*

Reply:

We thank the reviewer for raising this point, we revised the Discussion accordingly. Furthermore, prompted by the reviewers' questions (see also question 2 by the reviewer 1), we included a new Supplementary Note 4, where we analyze the connection between information-limiting correlations and noise correlations induced by On-Off dynamics. In our model, assuming that stimulus does not change the statistics of On-Off dynamics and that changes in the On-Off dynamics do not affect stimulus tuning, we found that the On-Off dynamics influence the strength of differential correlations and thus affect the saturation level of information. Specifically, the linear Fisher information is monotonically decreasing with the correlation length. Hence, a reduction of the correlation length leads to an increase in stimulus information, as we observed in superficial cortical layers during attention. However, the On-Off dynamics and stimulus tuning are likely interdependent in the cortical circuitry, where both arise from the same structured connectivity. In future work, deeper understanding of how On-Off fluctuations impact sensory coding will be possible using models with connectivity that supports stimulus tuning in addition to spatial receptive fields.

6) *I was a bit thrown by the use of 'On/Off' as a term for the two states. In fact, in the 'Off' state, the measured neuronal responses are still appreciable. This is in contrast to a rich literature on On/Off states (work of McCormick and others) that has shown almost total quiescence in the Off state. It might be less confusing (with respect to this prior literature) to refer to the effects here as 'Low' and 'High rate' states. At the very least, if the*

authors stick to the current terminology, they should explain that this same term means something quite different in other systems/network neuroscience literature.

Reply:

We clarified in Discussion the use of the terms “On-Off” dynamics versus “Up-Down” states (see also question 2 by the reviewer 2). Up-Down states were originally used to refer to the two modes of the bimodal distribution of membrane potentials, and now are also used to refer to spiking (Up) and silent (Down) phases of population activity during slow-wave sleep and anesthesia [Harris & Thiele, Nat. Rev. Neurosci. 12, 509 (2011)]. On-Off dynamics refer to spontaneous transitions between episodes of vigorous (On) and faint (Off) spiking that occur synchronously across layers in the cortex of behaving monkeys. Different terms “On-Off” versus “Up-Down” dynamics help to clarify that these phenomena are observed in different behavioral states (awake versus sleep/anesthesia). We introduced the term On-Off dynamics in our previous published work [Engel et al, Science 354,1140 (2016); van Kempen et al, Neuron 109, 894 (2021)], where we first observed this phenomenon in behaving animals. We would therefore like to continue using this term for the consistency with our previous work.

REVIEWERS' COMMENTS

Reviewer #1 (Remarks to the Author):

The authors have appropriately addressed all my comments.

Reviewer #3 (Remarks to the Author):

The authors have addressed the majority of my concerns. I support publication.